# Finding the rhythm: Humans exploit nonlinear intrinsic dynamics of compliant systems in periodic interaction tasks

**Annika Schmidt**[1,2,3]*, **Marion Forano**[3,4], **Arne Sachtler**[1,2], **Davide Calzolari**[1,2], **Bernhard M. Weber**[2], **David W. Franklin**[3,4], **Alin Albu-Schäffer**[1,2,3]

**1** Sensor Based Robotic Systems and Intelligent Assistance Systems, TUM School of Computation, Information and Technology, Technical University of Munich (TUM), Garching, Germany, **2** Institute of Robotics and Mechatronics, German Aerospace Center (DLR), Wessling, Germany, **3** Munich Institute of Robotics and Machine Intelligence (MIRMI), Technical University of Munich (TUM), Munich, Germany, **4** Neuromuscular Diagnostics, TUM School of Medicine and Health, Technical University of Munich (TUM), Munich, Germany

* annika.schmidt@dlr.de

**Data Availability Statement:** The recorded experiment and simulation data as well as the code to analyze the data and produce the figures

## Abstract

Activities like ball bouncing and trampoline jumping showcase the human ability to intuitively tune to system dynamics and excite motions that the system prefers intrinsically. This human sensitivity to resonance has been experimentally supported for interactions with simple linear systems but remains a challenge to validate in more complex scenarios where nonlinear dynamics cannot be predicted analytically. However, it has been found that many nonlinear systems exhibit periodic orbits similar to the eigenmodes of linear systems. These nonlinear normal modes (NNM) are computable with a recently developed numerical *mode tool*. Using this tool, the present resarch compared the motions that humans excite in nonlinear systems with the predicted NNM of the energy-conservative systems. In a user study consisting of three experiment parts, participants commanded differently configured virtual double pendula with joint compliance through a haptic joystick. The task was to alternately hit two targets, which were either aligned with the NNM (Experiments 1 and 2) or purposefully arranged offset (Experiment 3). In all tested experiment variations, participants intuitively applied a control strategy that excited the resonance and stabilized an orbit close to the ideal NNM of the conservative systems. Even for increased task accuracy (Experiment 2) and targets located away from the NNM (Experiment 3), participants could successfully accomplish the task, likely by adjusting their arm stiffness to alter the system dynamics to better align the resonant motions to the task. Consequently, our experiments extend the existing research on human resonance sensitivity with data-based evidence to nonlinear systems. Our findings emphasize the human capabilities to apply control strategies to excite and exploit resonant motions in dynamic object interactions, including possibly shaping the dynamics through changes in muscle stiffness.

presented in this manuscript are available on figshare under the link: https://doi.org/10.6084/m9.figshare.24032679.

**Funding:** A.S. and M.F. were partly funded by the TUM Integrative Research Fund, provided by the seed funding initiative of the Munich Institute of Robotics and Machine Intelligence (MIRMI). The research was additionally supported by the European Research Council (ERC) through the European Union's Horizon 2020 Research and Innovation Programme under Grant 835284. The funders had no role in study design, data collection and analysis, decision to publish, or preparation of the manuscript.

**Competing interests:** The authors have declared that no competing interests exist.

## Author summary

Without thinking about it, humans intuitively excite resonant motions in everyday object interactions, despite the complex and nonlinear nature of their dynamics. Computing these nonlinear dynamics is challenging, but it is essential to verify if the excited object motion matches the objects' intrinsic dynamics. Using a new numerical tool, we could predict these intrinsic dynamics. In a human user study, participants were tasked with exciting a virtual double pendulum through a haptic joystick. The excited motions were then compared to the intrinsic nonlinear dynamics predicted by the tool. The experiments verified that participants intuitively excited the resonance frequency of the nonlinear system and stabilized motion trajectories close to the computed intrinsic ones. Experimental variations also indicated that humans shape the system dynamics by changing their arm stiffness to create resonances that better align with the task. These findings support existing research showing that humans are highly sensitive to resonance and exploit it intuitively for tasks when possible.

## Introduction

Humans exhibit remarkable dexterity and versatility while using their upper limbs for daily activities. We can easily handle countless different objects, even when they are complex in shape, flexible in material, have multiple degrees of freedom (DOFs), or exhibit highly nonlinear or even chaotic dynamic behavior. In fact, studies show that humans can partially follow [1] and learn to predict [2, 3] chaotic system behavior. It is likely that the Central Nervous System (CNS) also employs chaotic control to explore system dynamics and tune into intrinsic motion patterns to excite resonant behavior [4]. This enables orchestrating numerous DOFs together, reducing the needed control effort. Everyday examples showcase human intuitive resonance sensitivity: without conscious effort, humans induce oscillatory motions when jumping on a trampoline or bouncing a ball by tuning to the intrinsic system dynamics [5, 6]. This excites the resonant frequency, which appears to be more predictable for humans [7, 8]. Predictability facilitates forming an internal model of objects and their dynamic behavior in interactions [9–12]. By learning and internalizing physical object models, humans can anticipate system motions, enabling *anticipatory control* strategies that leverage this knowledge [13–15]. Experiments have suggested that predictability becomes a priority for human control in rhythmic movement tasks. They are willing to sacrifice metabolic efficiency by adopting control trajectories that yield higher reaction forces and are less smooth [16, 17]. Instead of applying precise force control, humans tune their hand impedance to the system to take advantage of interactive dynamics [18, 19]. A cup balancing experiment showed that participants always chose similar starting positions to initialize the task [20], proving that humans can get an intuitive feeling even of complex nonlinear system dynamics.

Despite daily encounters with nonlinear dynamics, empirical evidence of human resonance sensitivity is mostly limited to interactions with simple systems exhibiting linear dynamics [21, 22]. Testing if humans intuitively excite and stabilize intrinsic nonlinear behaviors requires nonlinear dynamics to be computable in advance for comparison. But unlike the eigenmodes of linear systems [23, 24], intrinsic motions in nonlinear systems cannot be derived analytically. Instead, periodicity in chaotic systems can be found and stabilized through phase synchronization [2, 25], but multiple possible periodic orbits make it hard to predict which solutions emerge in human interactions. However, recent research showed even

nonlinear systems like the double pendulum, a classic example of chaotic behavior [26], display predictable, computable periodic orbits [27, 28]. These orbits can be determined with numerical methods based on algebraic topology and differential geometry. One class of these periodic orbits are nonlinear normal modes (NNM), similar in appearance and description to the linear analogy of eigenmodes. Analogously, they oscillate with a characteristic eigenfrequency between two turning points, where all velocities are zero. However, unlike linear eigenmodes, the shape of the nonlinear mode and the associated eigenfrequency changes with increasing energy levels and does not always cross the system's equilibrium. Yet, NNMs allow us to predict intrinsically preferred motions for conservative nonlinear systems on defined energy levels computable with a *mode tool* developed by our group [29, 30]. This tool enabled us to investigate whether humans are indeed sensitive to resonance in nonlinear systems and intuitively excite and stabilize a system close to its NNM.

We directly test this hypothesis with a human user study, where participants had to excite periodic motions in a virtual compliant double pendulum. Through a haptic joystick, participants moved a virtual *motor link* coupled to the first link $l_1$ of the double pendulum by a spring $k_1$ (Fig 1A and 1B). Rotating the motor link induced motions in the pendulum system and the spring forces were reflected to the user as haptic force feedback. By varying the equilibrium of the spring $k_2$ connecting the first and second pendulum link, different pendulum configurations (*P0, P90, P45*) could be tested (Fig 1C). In each case, the task was to alternately hit two

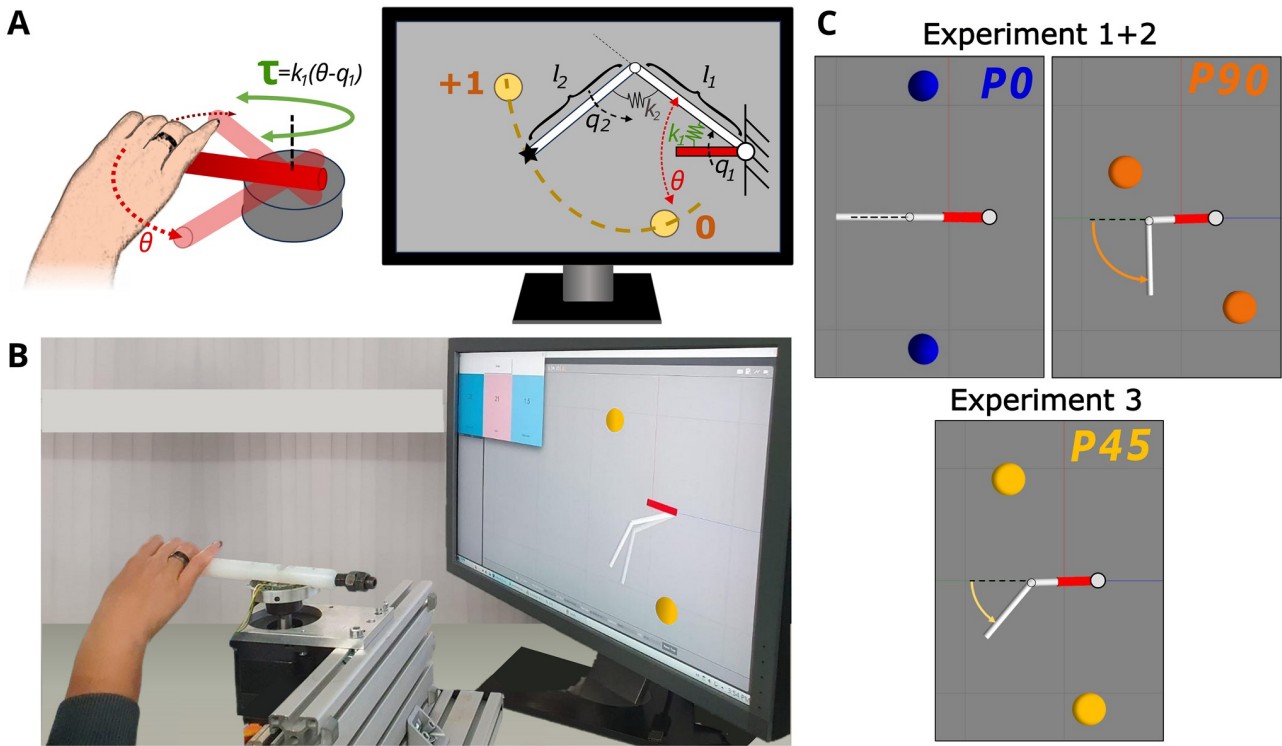

**Fig 1. Experimental setup.** (A) The participants rotate a haptic joystick that maps 1:1 to the position $\theta$ of the red virtual motor link on the screen. This link is coupled to the first link $l_1$ of the virtual compliant double pendulum by a spring $k_1$. Moving the joystick, and thus the virtual motor link, causes a deflection of the spring, which in turn induces the pendulum motion. The spring torque $\tau$ is reflected to the user as force feedback, rendered in green. The task is to hit two colored target balls with the second link endpoint (indicated by the star shape) as often as possible in 40 s. When the target is hit, the participant is rewarded with a point. If the pendulum does not reach the target or swings through it, no point is added to the score. (B) Participants can freely arrange themselves in front of the haptic joystick and position their arm to their preference. The joystick and motor link are both arranged to point to the left. (C) Three pendulum configurations are tested in the experiments, where the equilibrium position of the spring between the first and second link differs as indicated by the colored arrows: *P0*: $\boldsymbol{q}_{\mathrm{eq}} = (0, 0)^\circ$, *P90*: $\boldsymbol{q}_{\mathrm{eq}} = (0, 90)^\circ$, *P45*: $\boldsymbol{q}_{\mathrm{eq}} = (0, 45)^\circ$.

target balls to collect *hit* points. There were two major reasons for choosing a compliant double pendulum for the investigation. First, although the system is not inherently chaotic if the compliance is chosen high enough, the dynamics of the double pendulum is strongly nonlinear. It is the simplest model approximation of a flexible object, such as a whip or rope. As such, it can be easily extended with further links in future work to analyze interactions with more complex dynamics. Second, human resonance sensitivity is presumably not only important in object interactions but also for the control of one's own body. Hence, the compliant double pendulum can also be regarded as the simplest approximation of a human limb. Insights about how humans excite and stabilize such dynamics might support further investigations on human motor learning and control.

To identify the pattern underlying the human control approach, prior to the user experiment, three *baseline strategies* (*BL1–3*) were defined based on literature-reported human control principles. In simulations, each strategy commanded a position signal to the pendulum's motor link (Fig 1, red link), which was identical to what participants controlled by moving the haptic joystick. The amplitude of all strategies was empirically tuned to reach the two targets with the second pendulum link tip. The first strategy *BL1* was in line with our hypothesis that human resonance sensitivity extends to non-linear systems. Thus, *BL1* commanded a sine wave with the predicted eigenfrequencies of the different pendulum configurations to the motor link. Alternatively, the strategies *BL2* and *BL3* characterize strategies that do not excite resonance but resemble other known human control principles. *BL2* was inspired by findings that humans make use of motion constraints to reduce their effort [31], prioritizing the smoothness of hand and actuating forces [32]. Thus, we expected a strategy that would avoid extensive spring deflections and rapid direction changes to ensure low forces and smooth curves. Hence, *BL2* commanded a sine wave at a frequency much lower than the predicted NNM frequencies such that motor link speed synchronizes with both pendulum links and avoids spring deflections. Finally, *BL3* was a bang-bang controller. This strategy can model arm reaching motions following a minimum acceleration with constraints principle [33] and match the muscle activations in this motion [34]. Previous studies also suggested this approach to excite eigendynamics [35], later adapted for robot applications [36]. This robotic controller was used to model *BL3* in the experiments.

To quantify the participant performance and compare it with *BL1–3*, four metrics were defined. First, the achieved target hit scores were determined as a measure of successful task execution and precision. Second, the oscillation frequency $f_{osc}$ of the excited pendulum was assessed as an apparent indicator of resonant behavior. Third, we introduce the *mode metric $\eta$* to quantify how close the excited pendulum motions through the participants and *BL1–3* were to the respective NNM. It was expected that the pendulum motions in the experiment would not congruently match the ideal NNM path, since the NNM computation can only be performed for energy-conservative systems [27]. However, in the experimental settings, joint friction had to be added to the pendulum models to require a control input to be evaluated. According to our hypothesis, humans would intuitively stabilize the damped system on a trajectory close to the NNM if the intrinsic dynamics was exploited. Thus, the lower the mode metric value $\eta$, the closer the excited pendulum motion to the predicted NNM trajectory. The metric was derived from the time-independent method of *Dynamic Time Warping (DTW)* [37, 38] to allow joint trajectory comparisons for varying oscillation frequencies. Finally, we juxtapose the joystick motion translating to the motor link trajectory $\theta$ to the pendulum link paths $q_1$. Specifically, the deflection ratio $\rho = \max(\theta)/\max(q_1)$ and phase lag $\phi$ between these two links were analyzed since a high amplitude difference and a lag value of $\frac{\pi}{2}$ characterize resonant behavior in linear systems [39].

## Results

### Experimental setup and variations

For each of the tested pendulum configurations *P0*, *P90* and *P45* (Fig 1C), the respective NNM was computed to characterize the intrinsically preferred system motions for the conservative case. For details on the NNM computation, refer to the method section. For the pendulum configurations *P0* and *P90*, the locations of the targets balls that had to be hit were determined by the turning points of the NNM trajectories taken as reference (Fig 1C). For *P45*, the targets were purposefully arranged off the NNM trajectory. Each *hit* of the targets was rewarded with a point, motivating the participants to achieve a high score. No point was awarded if the second pendulum link did not reach the targets (*undershoot*) or swung through it (*overshoot*).

Three individual experiments were carried out, each focusing on a different research aspect. Experiment 1 should identify the underlying human control strategy to excite nonlinear resonance in the *P0* and *P90* pendulum, where targets aligned with the NNM. Each pendulum configuration appeared four times in random order. In each trial, the participant had 40 s for the task to achieve as many hit points as possible. Experiment 2 assessed the robustness of the identified strategy by dividing participants into two groups: one with decreased target radius ($\downarrow r_t$) and one with increased pendulum link mass ($\uparrow m_2$). The task and trial time remained the same, but participants were informed of the changes and had a short break to recover. Still, each pendulum configuration appeared four times in random order. Finally, Experiment 3 tested with the third pendulum configuration *P45* if and how the human control strategy would change when the targets were not located on the NNM, i.e., not aligned to the intrinsic system trajectory. To avoid biasing the participants' control strategy, they were left unaware whether the targets were co-located with the NNM (Exp. 1) or not (Exp. 3). Thus, the trials of Experiment 3 were randomly shuffled in with the trials of Experiment 1, appearing as a third pendulum configuration with the same task and trial time. This also aided to limit learning effects by avoiding to present one pendulum configuration exclusively in sequence.

### Experiment 1

Experiment 1 tested whether human resonance sensitivity extends to nonlinear dynamics, such that they intuitively excite intrinsically preferred motions. Using the introduced metrics, the motions excited by the participants ($n = 20$) over all encountered trials in the *P0* and *P90* pendulum were compared to the performance of *BL1–3* to identify the approach underlying the human controller.

**Task success.** While the individual hit score was of secondary interest, it validated the participants' overall ability to complete the task successfully. In both pendulum configurations *P0* and *P90*, most swings resulted in a counted hit with comparatively few swings having too much (*overshoot*) or too little (*undershoot*) energy (Tables 1 and 2). On average, participants scored 158 ± 15 hits with the extended *P0* pendulum, achieving an 82% accuracy relative to the ideally achievable score of the NNM. With the flexed *P90* configuration, participants averaged 209 ± 9 hits, translating in an even higher accuracy of 92%. The maximally achievable hit points with the resonance frequency was 192, while *BL3* could achieve 241 hits and *BL2* only 99 hits. The fact that the participants were closest to this score, even though a higher number would have been achievable with *BL3*, could suggest an initial similarity between the participant approach and *BL1*.

**Oscillation frequency.** According to our hypothesis, we expected that humans would intuitively excite and stabilize the pendulum systems close to their respective eigenfrequencies $f_{\text{res}(P0)} = 0.78$ and $f_{\text{res}(P90)} = 0.93$Hz, which was thus used for *BL1*. The average oscillation

**Table 1. Overview of the evaluated metrics for the *P0* configuration.**

| *P0* | [Hz] $f_{osc}$ | mode metric $\eta$ | deflection ratio $\rho = \frac{\max(\theta)}{\max(q_1)}$ | phase lag $\phi$ | hits | over-shoot | under-shoot |
|---|---|---|---|---|---|---|---|
| NNM | 0.78 | 0 | 0 | - | 192 | - | - |
| *BL1* | 0.78 | 13.46 | 0.11 | $0.43\pi$ | 192 | - | - |
| *BL2* | 0.39 | 38.38 | 0.72 | 0 | 99 | - | - |
| *BL3* | 0.98 | 40.37 | 0.55 | $0.86\pi$ | 241 | - | - |
| **Exp.1** ($n = 20$) | 0.77 ($\pm$ 0.03) | 14.34 ($\pm$ 1.64) | 0.14 ($\pm$0.04) | $0.43\pi$ ($\pm$0.16$\pi$) | 158 ($\pm$ 15) | 9 ($\pm$ 7) | 12 ($\pm$ 9) |
| **Exp.2** ($\downarrow r_t$) ($n = 10$) | 0.75 ($\pm$0.03) | 14.92 ($\pm$1.64) | 0.15 ($\pm$0.04) | $0.29\pi$ ($\pm$0.13$\pi$) | 139 ($\pm$19) | 15 ($\pm$12) | 15 ($\pm$5) |
| **Exp.2** ($\uparrow m_2$) ($n = 10$) | 0.53 ($\pm$0.02) | 6.14 ($\pm$1.77) | 0.14 ($\pm$0.04) | $0.58\pi$ ($\pm$0.22$\pi$) | 110 ($\pm$7) | 7 ($\pm$4) | 5 ($\pm$4) |
| NNM($\uparrow m_2$) | 0.52 | 0 | 0 | - | 131 | - | - |

**Table 2. Overview of the evaluated metrics for the *P90* configuration.**

| *P90* | [Hz] $f_{osc}$ | mode metric $\eta$ | deflection ratio $\rho = \frac{\max(\theta)}{\max(q_1)}$ | phase lag $\phi$ | hits | over-shoot | under-shoot |
|---|---|---|---|---|---|---|---|
| NNM | 0.93 | 0 | 0 | - | 228 | - | - |
| *BL1* | 0.93 | 28.58 | 0.14 | $0.39\pi$ | 228 | - | - |
| *BL2* | 0.46 | 54.61 | 0.73 | 0 | 117 | - | - |
| *BL3* | 1.21 | 73.27 | 0.7 | $0.77\pi$ | 299 | - | - |
| **Exp.1** ($n = 20$) | 0.92 ($\pm$ 0.04) | 22.19 ($\pm$ 2.07) | 0.17 ($\pm$ 0.03) | $0.44\pi$ ($\pm$0.16$\pi$) | 209 ($\pm$ 9) | 2 ($\pm$ 2) | 6 ($\pm$ 8) |
| **Exp.2** ($\downarrow r_t$) ($n = 10$) | 0.87 ($\pm$0.06) | 24.17 ($\pm$3.99) | 0.20 ($\pm$ 0.07) | $0.28\pi$ ($\pm$0.15$\pi$) | 190 ($\pm$13) | 4 ($\pm$4) | 9 ($\pm$8) |
| **Exp.2** ($\uparrow m_2$) ($n = 10$) | 0.65 ($\pm$0.06) | 27.22 ($\pm$6.54) | 0.17 ($\pm$ 0.04) | $0.59\pi$ ($\pm$0.16$\pi$) | 149 ($\pm$7) | 2 ($\pm$2) | 4 ($\pm$4) |
| NNM ($\uparrow m_2$) | 0.64 | 0 | 0 | - | 160 | - | - |

frequency participants excited was 0.77 ± 0.03 Hz and 0.92 ± 0.04 Hz for *P0* (Table 1) and *P90* (Table 2), respectively. The statistical comparison showed no significant difference to the ideal $f_{res}$ (*P0*: $t(19) = -1.23$; $p = .23$, *P90*: $t(19) = -1.20$; $p = .25$), indicating that the participants intuitively matched the pendulum eigenfrequencies. Comparing with *BL2* and *BL3* showed strongly significant differences with all *p*-values $p < 0.001$ (Table C in S1 Table).

**Mode metric.** Although it was possible to hit the targets with all tested baseline strategies (Fig 2, left), differences in the excited trajectories become apparent in joint space (Fig 2, middle and right). The mode metric quantifies these differences by computing the distance to the ideal NNM, as summarized in Tables 1 and 2. In a conservative case with a static motor link and one of the target locations as the initial position, the mode metric $\eta$ would be zero. However, since friction was added in the experimental setup to require control, no exact overlay with the ideal NNM was expected, neither for the participant data nor the baseline strategies. Instead, a low mode metric suggests closer proximity of the excited pendulum motion to the ideal NNM.

For the *P0* configuration, the mode metric resulted in $\eta(P0) = 14.34 \pm 1.64$. For the *P90* configuration, $\eta(P90) = 22.19 \pm 2.07$ was calculated. Comparing these values with the mode

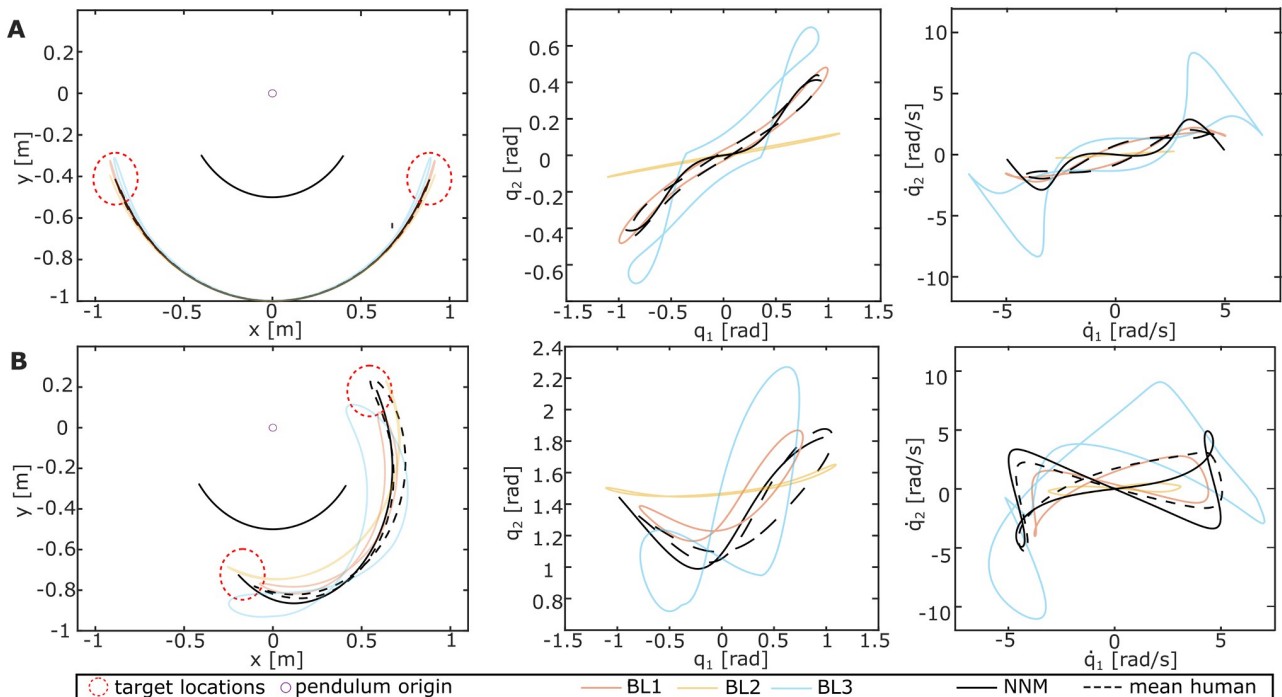

**Fig 2. Differently excited pendulum motions.** Compared are the pendulum motions excited by the baseline strategies *BL1–3* with the system's conservative NNM (solid black) and the averaged participant data (dashed black) for the (A) *P0* and (B) *P90* pendulum configuration. The resulting motions are shown in Cartesian space (left) and joint space, comparing positions (middle) and velocities (right).

metric of *BL1–3* (Tables 1 and 2) shows that the averaged participant data and *BL1* achieved the lowest values. This indicates that these two approaches excited pendulum motions closest to the conservative NNM, which becomes visually apparent when overlaying the participant data with *BL1–3* in Fig 2 (red and dashed black). To test this similarity statistically, the difference of the $\eta$-values between the participant data and *BL1–3* was compared against zero. It shows no significant difference for $\eta(P0)$ ($t(19) = 2.05$; $p = .053$). However, $\eta(P90)$ indicates a significant difference between the two strategies ($t(19) = -13.4$; $p < .001$). Thus, although for *P0*, we cannot reject the hypothesis that *BL1* underlies the human strategy, the difference in *P90* could suggest that the human controller might be more sophisticated than the *BL1*-sine to excite intrinsic system dynamics. Comparing the difference between the participants' $\eta$-values and *BL2–3* against zero shows strongly significant differences (*all*: $p < 0.001$, Table C in S1 Table). Thus, it can be rejected that participants apply an excitation strategy similar to *BL2–3*.

**Handle motion.** In the conservative case, the NNM characterizes a motion that the systems intrinsically follow when deflected from the correct initial position. This assumes the motor link is static at $\theta = 0$. In the experiment, friction was added to the system so that the participants had to continuously inject energy into the system by moving the motor link. Nevertheless, if the intrinsic dynamics are exploited, the motor link amplitude should remain small compared to the pendulum deflection. Plotting $\theta$ against $q_1$ for *P0* and *P90* over one period (from target to target) validated this expectation (Fig 3, left). On average, the first pendulum link $q_1$ moved $1.81 \pm 0.04$ rad $= 103.73 \pm 2.74°$, while the motor link $\theta$ had a maximum deflection of $0.26 \pm 0.06$ rad $= 15.12 \pm 3.47°$ for *P0*. Similar differences between $q_1$ ($1.88 \pm 0.11$ rad $= 107.54 \pm 6.50°$) and $\theta$ ($0.32 \pm 0.06$ rad $= 18.71 \pm 3.49°$) show for the *P90* configuration. Relatively, this means that the motor link was only moved 14% and 17% of the first pendulum

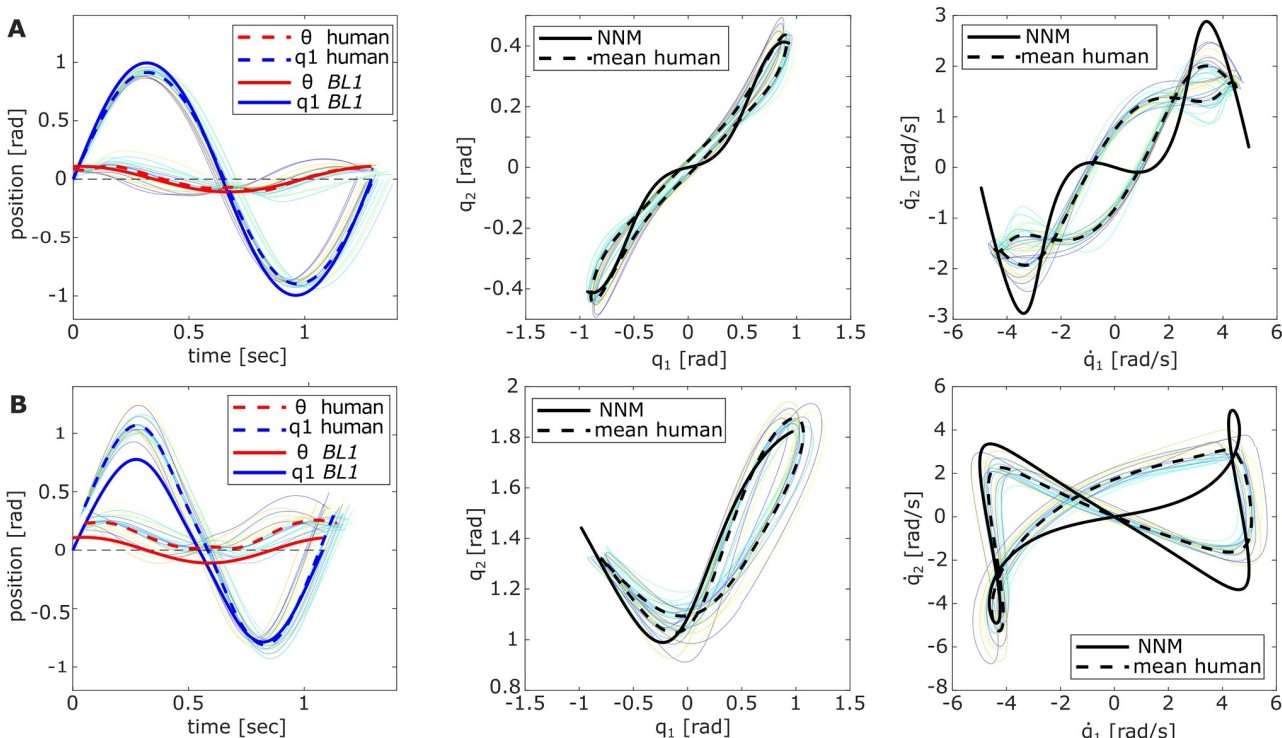

**Fig 3. Overlay of excited pendulum trajectories in Exp. 1.** Each light-colored line corresponds to the trial average of an individual participant with (A) *P0* and (B) *P90*. The left plot visualizes the applied joystick motion, i.e., motor link $\theta$ (red) and first pendulum link $q_1$ (blue) over one period (target to target) comparing the averaged participant data (dashed) to *BL1* (solid). The middle and right plots show the respective trajectories in position and velocity space compared to the NNM trajectory.

link trajectory for *P0* and *P90*, respectively. The observations were similar for *BL1* applying a sine wave with the computed resonance frequencies. The trajectory length of the motor link was 11% and 14% of the path traveled by the first pendulum link for *P0* and *P90*, respectively. Both *BL2* and *BL3* had to deflect the motor link much more to achieve a pendulum amplitude that hit the targets (Tables 1 and 2). Thus, the participant strategy appears closest to *BL1*, but the *P90* configuration reveals a difference: Instead of oscillating the pendulum around the zero-equilibrium of the first spring $k_1$, the handle oscillated around −0.13 rad = −7.44°, thus changing the first entry of the equilibrium position $\boldsymbol{q}_{\mathrm{eq}}$ (Fig 3B, left).

The averaged participant data indicated an overall phase lag of $0.43\pi \pm 0.16\pi$ for *P0* and $0.44\pi \pm 0.16\pi$ for *P90* (Fig 3). Comparing these values to the baseline strategies (Tables 1 and 2), they show no significant differences to *BL1* (*P0*: $t(19) = -0.16$; $p = .88$, *P90*: $t(19) = 1.30$; $p = .21$), while *BL2–3* both are significantly different (*all*: $p < 0.001$, Table C in S1 Table). Nevertheless, the large standard deviations indicate that the individually applied phase lag varied widely. For the extended pendulum *P0*, the individual phase lags were found to be between $0.15\pi$ and $0.70\pi$, while in the flexed *P90* configuration, the values varied between $0.21 - 0.74\pi$. To investigate whether the individually chosen phase lag of the participants influenced their performance, we relate it to the achieved hit score (Fig 4A) and the determined mode metric value per participant (Fig 4B). Comparing the phase lags with the respective hit scores shows positive trend lines for both pendulum configurations (Fig 4A). Computing the Pearson correlation between the two variables shows indeed a significant correlation for *P90* ($r(18) = .59$; $p = .007$), but no significance for *P0* ($r(18) = .37$; $p = .10$). The comparison with the mode metric

 

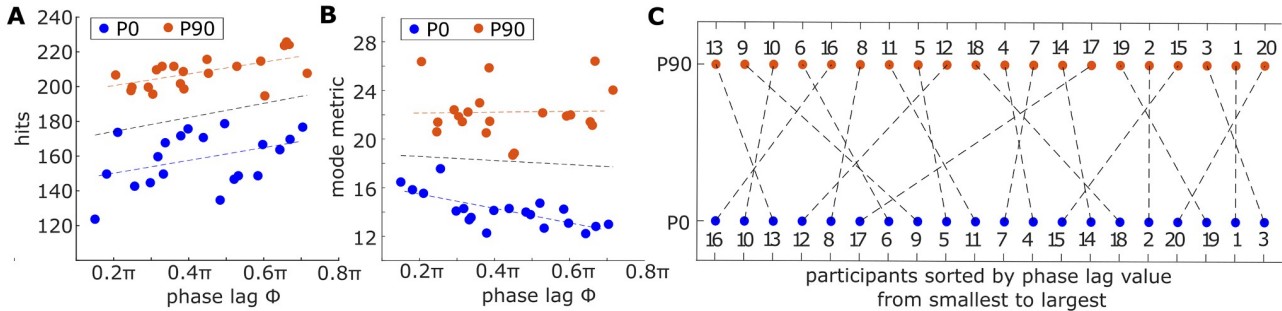

**Fig 4. Trend comparison of individual performance metrics.** The individually applied phase lag between the input motor link and the output pendulum link per participant are correlated with (A) the obtained hit score and (B) mode metric values. (C) Sorting the phase lag values per participant from smallest to largest reveals that participants remained consistent with their chosen strategy for the two tested pendula.

shows a significant correlation for *P0* ($r(18) = -.66$; $p = .001$), indicating that larger phase lags might achieve lower $\eta$-values. This significance could, however, not be found for the *P90* pendulum ($r(18) = .13$; $p = .58$). Sorting the individually applied phase lag per participant for the two configurations *P0* and *P90* (Fig 4C) reveals that all participants were relatively consistent in their excitation strategy, meaning the applied phase lags for both pendulum configurations were in a similar range per participant.

## Experiment 2

Two variations to the original experiment were applied to assess the consistency of the participants' control strategy. While for half of the participants, the target radius $r_t$ was decreased, the other half experienced an increased second link mass $m_2$ ($n = 10$).

Although decreasing the target radius led to a higher precision requirement, the participants' hits only slightly decreased for both pendulum configurations (Tables 1 and 2, Exp.2: ↓ $r_t$). The scores remained closest to the achievable *BL1* points and resulted in hit rates of 0.72% (*P0*) and 0.83% (*P90*) compared to the ideal NNM motion. The associated frequencies of the excited pendulum oscillations also slightly decreased to 0.75 ± 0.03 Hz and 0.87 ± 0.06 Hz for *P0* and *P90*, respectively. Comparing with the ideal eigenfrequencies applied by *BL1* showed significant differences for both configurations (*P0*: $t(9) = -4.44$; $p = .002$, *P90*: $t(9) = -3.58$; $p = .006$). Interestingly, no significant differences were found with *BL1* for the mode metric in both pendulum configurations, suggesting that the excited pendulum trajectories were still close to the ideal NNM (Fig 5, middle and right). The overall handle motion to excite the systems also appeared consistent with the previous observations (Fig 5, left). The deflection ratio $\rho$ between $\theta$ and $q_1$ still indicated that the handle was moved little compared to the excited pendulum amplitude, but the averaged phase lag $\phi$ showed a slight decrease. Both metrics were significantly different for *P0*, but not for *P90*. Detailed statistics are reported in Table D in the S1 Table. Thus, the acquired metrics suggest that participants still excited intrinsic pendulum dynamics, but the difference in oscillation frequency and phase lag indicate a change in the control or system.

Increasing the second pendulum link mass $m_2$ in the second experiment variation alters the system dynamics, necessitating the recomputation of the respective NNM. The NNM frequencies of the altered conservative systems then shift to 0.52 Hz and 0.64 Hz for *P0* and *P90*, respectively (Tables 1 and 2, NNM(↑ $m_2$)). Consequently, the maximally achievable hit score of the NNM decreased. Compared to these new ideal scores, participants achieved hit

 

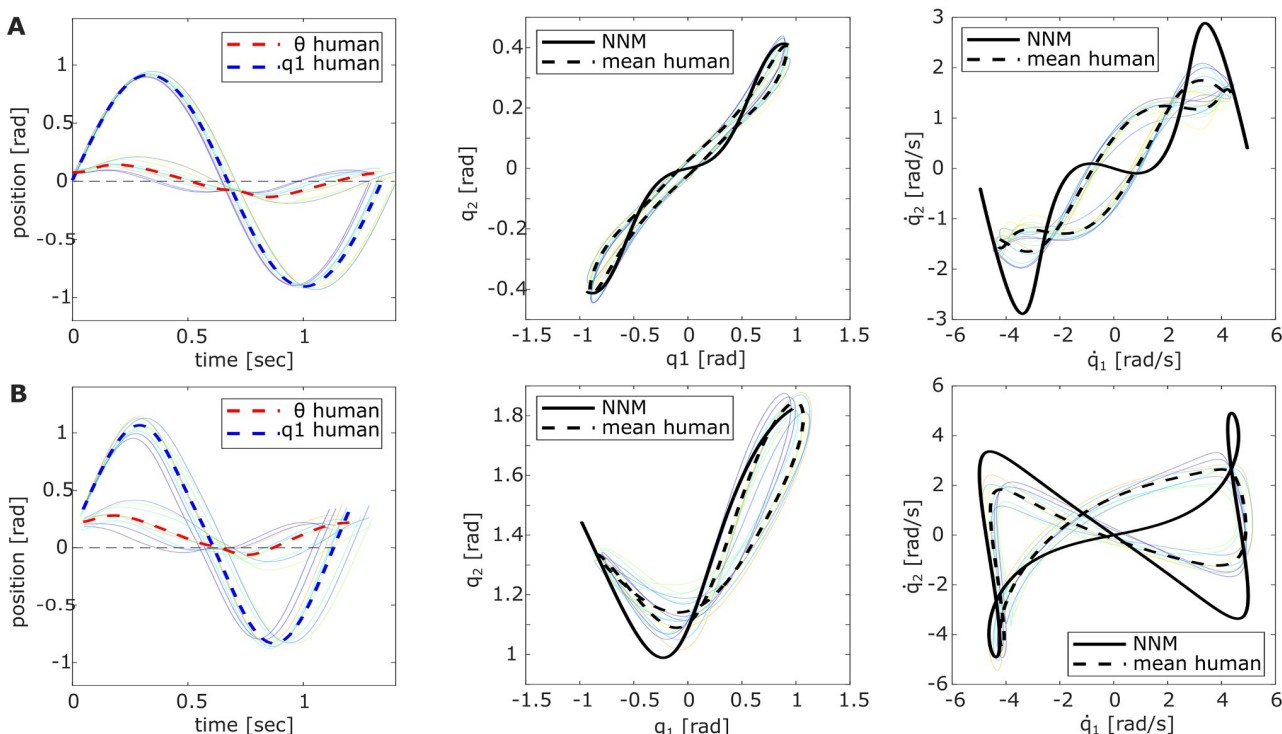

**Fig 5. Pendulum trajectories of Exp. 2 with decreased target radius (↓ $r_t$).** Light-colored lines show the averaged trials per individual participant and the overall averages are dashed for (A) *P0* and (B) *P90*. The left plots compare the red motor link motion $\theta$ to the first pendulum link $q_1$ (blue) per period. Middle and right plots compare the respective trajectories in position and velocity space with the NNM.

rates of 0.88% (*P0*) and 0.98% (*P90*), outperforming Experiment 1. As expected, the participants remained sensitive to the resonance frequencies and intuitively excited the pendulum systems with 0.53 ± 0.02 Hz and 0.65 ± 0.06 Hz. showing no significant differences with the ideal eigenfrequencies in *P0* and *P90* (Table D (red) in S1 Table). The mode metric comparing with the newly computed NNM validates that the participants excited the intrinsic dynamics similarly well as seen in Experiment 1. The very low $\eta$-value for *P0* even indicates that the stabilized orbit was closer to the ideal NNM (Fig 6). Again, the applied handle motion was similar in the relative amplitude differences for both pendulum configurations but varied in the phase lag $\phi$.

## Experiment 3

Experiments 1 and 2 demonstrated that the resonance sensitivity of humans extends to nonlinear systems, even when system or precision requirements change. However, in these experiments, the required oscillation task aligned with the intrinsically preferred pendulum motions, i.e., the targets coincided with the turning points of the ideal NNMs. Since such alignments are uncommon in everyday scenarios, we conducted an additional experiment with all participants (*n* = 20) to explore how humans excite a nonlinear system when the task and intrinsic dynamics are not aligned. This experiment was carried out with the *P45* pendulum, where $\boldsymbol{q}_{\text{eq}}$ = (0, 45)°. The target balls for this configuration did not coincide with the turning points of the system's NNM but were arranged on a radius between the *P0* and *P90* targets (Fig 1C, yellow).

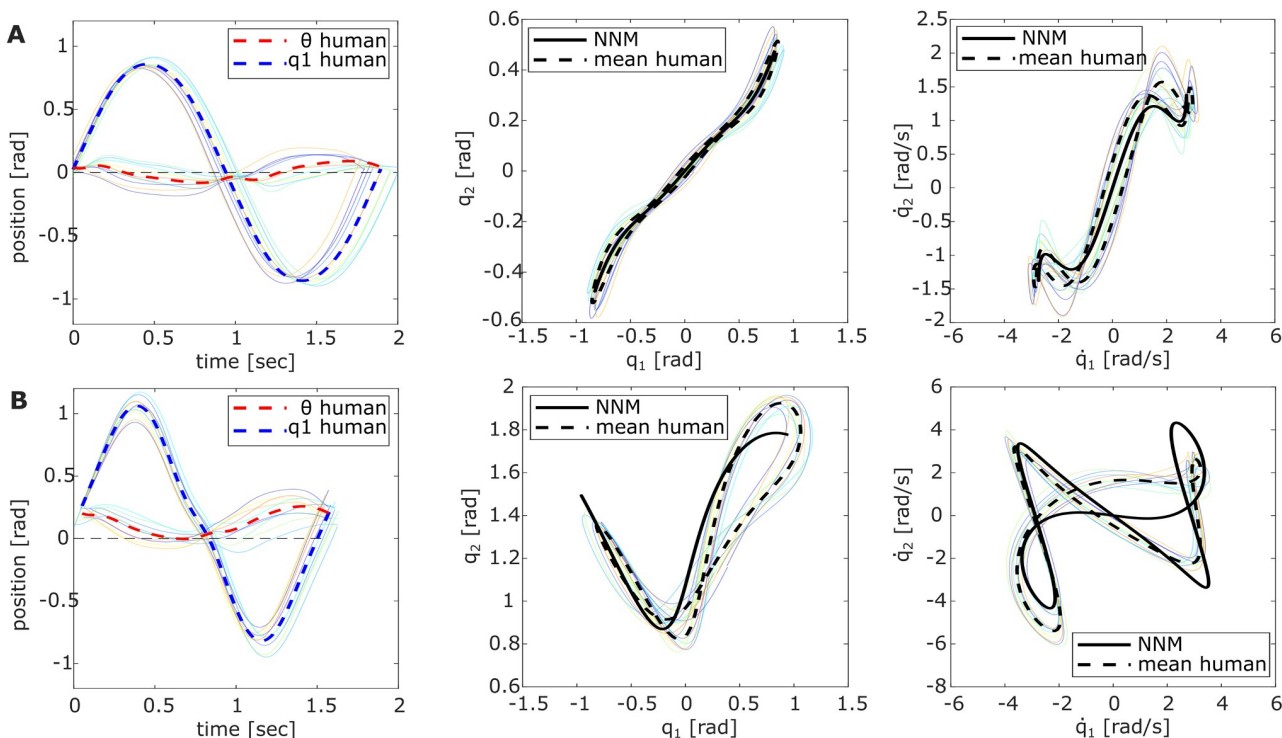

**Fig 6. Pendulum trajectories of Exp. 2 with increased link mass (↑ $m_2$).** Light-colored lines show the averaged trials per individual participant and dashed are the overall averages for (A) *P0* and (B) *P90*. The left plots compare the red motor link motion $\theta$ to the first pendulum link $q_1$ (blue) per period. Middle and right plots compare the respective trajectories in position and velocity space with the NNM.

Although the targets were not aligned with the NNM, the participants could still achieve a hit rate of 85%, thus lying in between the rates achieved with the *P0* and *P90* configurations. The participants intuitively excited an oscillation at 0.81 ± 0.02 Hz (Table 3), which was not significantly different from the determined NNM eigenfrequency of 0.82 Hz ($t(19) = -2.04$, $p = 0.06$). However, the mode metric of the participants and *BL1–3* indicate that the excited pendulum motion had a much larger distance from the ideal NNM trajectory, which was expected since the targets were not located on the turning points of the NNM (Fig 7, middle and right). The difference between the participants' $\eta$-values and *BL1–3* showed to be significantly different from zero for all cases (*all*: $p < 0.001$), indicating the participant strategy was

**Table 3. Overview of the evaluated metrics for the *P45* configuration.**

| *P45* | [Hz] $f_{osc}$ | mode metric $\eta$ | deflection ratio $\rho = \frac{\max(\theta)}{\max(q_1)}$ | phase lag $\phi$ | hits | over-shoot | under-shoot |
|---|---|---|---|---|---|---|---|
| NNM | 0.82 | 0 | - | - | 203 | - | - |
| *BL1* | 0.82 | 72.96 | 0.12 | $0.49\pi$ | 203 | - | - |
| *BL2* | 0.41 | 84.62 | 0.76 | 0 | 104 | - | - |
| *BL3* | 1.05 | 82.09 | 0.66 | $0.85\pi$ | 259 | - | - |
| **Exp. 3** ($n = 20$) | 0.81 (± 0.04) | 71.84 (± 1.33) | 0.14 (± 0.03) | $0.45\pi$ (±0.14$\pi$) | 172 (± 13) | 6 (± 5) | 11 (± 11) |
| NNM(↑ $k_1$) | 0.88 | 0 | - | - | 217 | - | - |

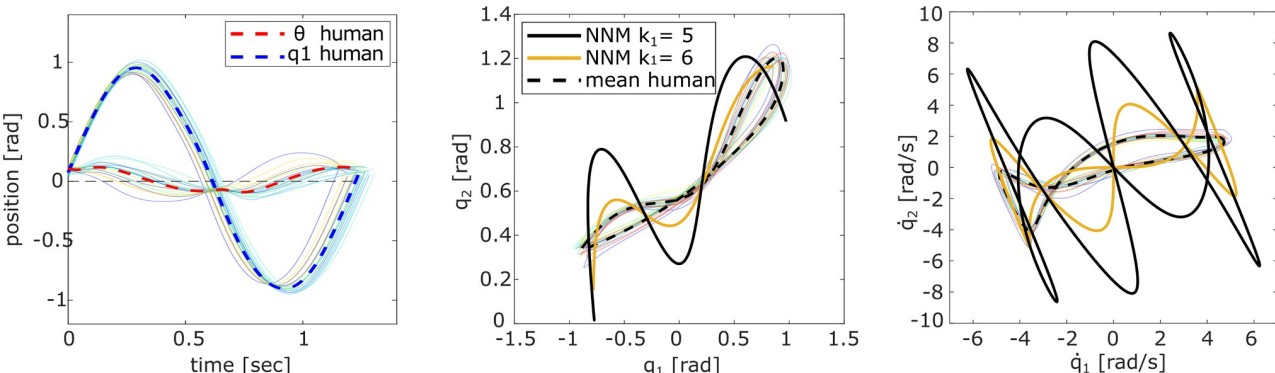

**Fig 7. Excited pendulum trajectories of Exp. 3 with the P45 configuration.** Light-colored lines show the average data over all trials per individual participant. The left plots compare the red motor link motion $\theta$ to the upper pendulum link coordinate $q_1$ (blue) cut by period (target to target). The middle and right plots show the respective trajectories in position and velocity space, compared to the NNM of the original pendulum system (black solid) and a system where the upper spring stiffness was altered to $k_1 = 6$ N m rad$^{-1}$ (yellow).

closer to the NNM than any baseline strategy. The applied motion to the motor link neverthe-less proposed that the overall control approach of the participants was consistent with the ones seen in the previous experiments, as the deflection ratio $\rho$ and the phase lag $\phi$ remained in a similar range as observed in Experiment 1 (Fig 7, left). Compared with *BL1*, no significant dif-ference was found for the phase lag ($t(19) = -1.25$, $p = 0.23$), suggesting the participant strategy is still most similar to *BL1*. All statistics are summarized in Table E in S1 Table.

In an attempt to understand how participants were still able to hit the targets, we consider that participants might have stiffened their wrist muscle tension influencing the upper spring $k_1$. To test this hypothesis, the original participant data is also compared with the recomputed NNM of an alternative *P45* pendulum with $k_1 = 6$ N m rad$^{-1}$, for which the turning points of the NNM are within or at least close to the target locations. This altered system's NNM appears closer to the excited pendulum motions of the participants (Fig 7), quantified by a lower mode metric of $\eta(\uparrow k_1) = 31.91 \pm 1.16$. This value is closer to the $\eta$-values of *P0* and *P90* (Tables 1 and 2). However, the computed eigenfrequency for this changed system would be 0.88 Hz, which is faster than the motions the participants excited.

## Discussion

Three experiments were performed to examine whether human resonance sensitivity extends to nonlinear systems. Participants interacted with a haptic joystick to command a virtual com-pliant double pendulum, exhibiting dynamics similar to a human limb. The task was to excite the pendulum systems to alternately hit two targets as often as possible. Three pendulum con-figurations were tested, determined by the set equilibrium positions of the second spring $k_2$. The excited pendulum motion could be compared to the intrinsic dynamics of the nonlinear system, characterized by the nonlinear normal mode (NNM). These NNMs describe periodic orbits that conservative nonlinear systems intrinsically follow when initial conditions are cor-rectly chosen. They can be determined with a newly developed *mode tool* [29, 30]. Investigat-ing the excited frequency and the distance from the intrinsic ideal NNM provided evidence that humans are sensitive to resonance even in nonlinear systems and exploit their inherent dynamics when possible.

## Human sensitivity to resonance and nonlinear normal modes

The pendulum motions excited by the participants support our hypothesis that humans are sensitive to resonance even in nonlinear systems. Intuitively, participants excited the numerically predicted eigenfrequencies of all pendulum configurations although a higher hit rate would have been achievable using control strategies driving faster oscillations, e.g., a bang-bang strategy *BL3* (Tables 1–3). Instead, all participants applied a continuous control input $\theta$ of small amplitude to the haptic joystick, which was overall most similar to *BL1*. At the correct frequency, this input excited resonant system behavior, indicated by the much larger output amplitude of the pendulum links compared to the input motor link. Moreover, not only was the resonance frequency matched, but the participants stabilized a periodic orbit close to the ideally predicted NNM (Fig 2) as quantified by the small values of the *mode metric*. The significant difference of this metric for *P90* could suggest that the human control signal is more sophisticated than the simple sine-wave commanded by *BL1*. Thus, it is a non-trivial observation that humans appear sensitive to resonance frequencies and the matching input shape alike, both of which appear important to stabilize intrinsic system motions.

Humans might intuitively excite systems at their resonance frequency, similar to *BL1* due to several factors. It has been shown that system behavior is easier to predict for humans when moving at its intrinsic frequency [7, 8]. Thus, exciting resonance likely requires less sensorimotor information processing [16], reducing the effects of sensorimotor delays [40]. This seems to make predictability a prioritized control objective during rhythmic movement tasks, even if this entails higher forces and less smooth trajectories [17]. The priority of predictability and stability [17, 20, 41] is supported by our experimental data by showing significant differences between the participant data and *BL2* and *BL3*. Although *BL2* would have led to smoother force curves due to the aligned motion of the motor and pendulum links, participants did not choose this strategy. Likewise, despite the higher hit score achievable with *BL3*, it did not characterize the participant control approach. Both of these possible control objectives seemed to be sacrificed for the more predictable resonant oscillations excited with a control approach most similar to *BL1* (Fig 2).

It needs to be mentioned that during the initial training period before the experiment, participants applied slower, more careful motions. This strategy showed some resemblance to the congruent motion pattern of *BL2*. However, this strategy was most likely not applied to minimize forces but to estimate the system dynamics of the interacted pendulum object. Since sufficient feedback was provided and we purposefully chose a system with dynamics familiar to humans, all participants could quickly adjust their internal system model [9–11, 42] and predict the system's behavior. After this initial training phase, the participants' strategies did not appear to improve or change further during the trials. Therefore, the effects of learning were not specifically investigated in this research but are expected to become more important in future studies when humans interact with objects with more unfamiliar behavior.

## Motor control strategies to excite nonlinear resonance

For the averaged participant data, the motor link amplitude and its phase lag to the pendulum links closely matched the input signal shape commanded by *BL1* (Fig 3, left). Both display low deflection ratios and phase lag values near $0.5\pi$, consistent with known input-output relationships seen when exciting resonant behavior in linear systems [39]. However, ideally, the first pendulum link was expected to always oscillate around its equilibrium position at $q_{eq_1} = 0$, which is realized when the mean $\theta$ position coincides with the defined $q_1$-origin (Fig 8A–8C). This expectation was met in the *P0* configuration, but the *P90* experiments revealed a slight shift of the mean motor link position towards the right ball for all participants (Fig 3B, left).

Recalculating the NNM for the shifted *P90*-equilibrium of $q_{eq} = (-7.44, 90)°$ shows neither a notable change of the eigenfrequency nor the shape of the NNM for the tested energy level. Therefore, the consistent handle shift among participants was most likely not applied to alter dynamics behavior but might have been triggered through visual cues. Visually, the flexed pose of the *P90* pendulum appears closer to the left target, although the second link tip is centrally aligned between both targets (Fig 1C, orange). This might have distorted the perceived symmetry of the participants and caused the slight handle shift to the right to compensate for this feeling. Future research should investigate the specific impact of visual cues on human abilities to excite nonlinear dynamics with the presented methods.

Another distinction between the participant data and *BL1* emerges when examining the individual *θ*-curves of the participants: While the phase lag of the averaged participant trajectory applied to the motor link aligned with that of the *BL1* input trajectory (Tables 1–3), the individually observed phase lag values varied between $0.2\pi$–$0.7\pi$ (Figs 3 (left) and 4). These values differ clearly from the optimal phase shift value of $0.5\pi$ that characterizes resonant behavior for a forced oscillation in linear systems [39]. Nevertheless, the consistently low deflection ratios and excited pendulum oscillations at the predicted eigenfrequencies across all participants indicate that resonant behavior was still triggered in all double pendulum configurations (Tables 1–3). This suggests that intrinsic elasticities of nonlinear systems can be exploited in more diverse ways, which aligns with recent research showing that resonance of nonlinear modes can be discovered for phase shifts different from $0.5\pi$ [43]. Although partly significant correlations between the phase lag and the achieved hit score or mode metric could suggest that the participant performance was dependent on the phase lag, the data is overall not strongly conclusive in this regard. For example, participant 2 applied with $0.6\pi$, a value close to the expected linear phase lag. Nevertheless, this participant achieved a comparatively low hit score and excited a pendulum motion further from the NNM than participants with similar lag values accomplished. In contrast, participant 5 applied a motor motion that only lagged $0.36\pi$ behind the pendulum motion and still had a relatively high hit score and low mode metric value. It is noticeable, however, that although the applied phase lag varied among the different participants individually, most users were consistent in their choice. This means that when the observed phase lag was low for the *P0* pendulum, it was in a similar value range for that same participant with *P90* (Fig 4C). Thus, it appears that no specific phase lag value was superior to excite resonance in the investigated nonlinear systems. At least for the regarded low friction case, the timing of energy injection into the system seemed to be of minor importance and did not determine the task performance. Instead, different phase lag values could successfully excite system resonance, suggesting that the applied strategy might depend more on the individual participant skills and possibly their internal system model.

Additionally, it appeared that the applied motor link motion of most participants shortly plateaued when the pendulum link crossed the equilibrium position $q_{eq_1} = 0$ (Fig 3). This indicates that the participants held the joystick handle steady for a moment when it was overtaken by the pendulum link, which was observable independent of the applied phase lag. With this feature, the individual control input of the participants appeared to be more complex than the simple sine wave shape of *BL1*.

It would be interesting to investigate further the influence of phase lag and the dedicated control signal shape in future research. To do so, systems with different levels of friction could be presented to the participants, so that the input trajectories need to be chosen more carefully to support intrinsic oscillations. Thereby, it might be possible to determine if an overall preferred phase lag emerges and if the human input curve might become more noticeably nonlinear.

## Human adjustments shaping resonance to accommodate tasks

The variations in Experiments 2 and 3 highlighted how robust the participants were in choosing and applying the already discussed overall control strategy similar to *BL1*.

Without added training time, participants accomplished the task in Experiment 2 when the pendulum system's second link mass $m_2$ was increased. Although the mass change altered the system behavior, the new intrinsic dynamics was intuitively excited, and the achieved hit rate was in the same range as with the initial parameters. For the second variation of Experiment 2 with decreased target size $r_t$, participants appeared consistent with their applied control strategy. The mode metric $\eta$ only increased slightly for both pendulum configurations, indicating the excited pendulum motions remained close to the predicted NNM path (Fig 5). However, the excited frequencies in *P0* and *P90* were slightly but significantly lower than the predicted eigenfrequencies. Moreover, the hit scores slightly decreased with comparatively more error attempts compared to the original target size of Experiment 1 (Tables 1 and 2: *overshoot, undershoot*).

To understand how the oscillation frequency could be lowered while the mode metric suggests still close proximity to the ideal NNM, the hand-joystick interaction of the participants is considered. The close match of the eigenfrequencies and low $\eta$-values in the previous experiments suggested that participants maximally stiffened their wrists when holding the joystick handle. This best mimics the idealized condition of the conservative systems, where the motor link was static at $\theta = 0$. However, changing the wrist stiffness could alter the coupled system dynamics [19]. Although, the human arm stiffness was not specifically considered in our experiment, a very rough approximation of such stiffness change can be made by assuming a changed stiffness for the first system spring $k_1$. Since the human arm and this spring act in series, it can be assumed that lowering the wrist stiffness would lower the overall stiffness seen by the first pendulum link, such that changing $k_1$ can be used to gain first insights. A more detailed modeling would exceed the scope of this paper, but will be considered for further research. Here, we simply recalculated the NNM of the pendulum systems with slightly decreased first spring stiffness $k_1 = 4.75$ N m rad$^{-1}$. This results in eigenfrequencies of $f_{\mathrm{res}(P0)} = 0.758$ Hz and $f_{\mathrm{res}(P90)} = 0.90$ Hz, showing no significant difference to the frequencies excited by the participants to reach the targets with decreased radius (*P0*: $t(9) = -1.55$; $p = 0.16$, *P90*: $t(9) = -1.94$; $p = 0.06$). Simultaneously, the NNMs trajectories with the altered stiffness only changed marginally, which explains why $\eta$ remained low. In this way, the participants could slow down the system and still exploit the intrinsic dynamics to hit the targets. Theoretically, this observation is counter-intuitive, as ideally matching the NNM should result in a perfect score as long as the targets are arranged on the turning points of the brake orbits. However, it is likely that participants intuitively preferred slower system oscillations for increased task difficulty as suggested by Fitt's Law [44]. Thus, by softening their wrist stiffness, participants might have subconsciously shaped the system dynamics to better fit their preferences. A similar method is also applied for robotic motion control. Based on the concept of controlled Lagrangian [45], the robot controller slightly changes closed-loop dynamics of the system to match desired tasks.

Similarly, participants might have shaped the system dynamics to complete the task in Experiment 3, where the turning points of the NNM and the target locations did not coincide. Computing the NNM with an altered spring stiffness $k_1 = 6$ N m rad$^{-1}$ could suggest further stiffening of the hand-joystick connection. In this way, the turning points of the altered NNM lie again within or at least close to the targets such that humans can better exploit the system elasticities. This is supported by the lowered $\eta$ value for the NNM of the altered system (Fig 7, yellow). However, examining the oscillation frequency excited by the participants contradicts

this idea (Table 3). Participants appeared to excite the eigenfrequency of the original NNM, not the one of the system with increased $k_1$ stiffness. Thus, further experiments, including EMG measurements, will be necessary to investigate this hypothesis explicitly.

Nevertheless, the experiments suggest that humans are not only sensitive to exciting the physically inherent system dynamics but might also intuitively shape the dynamics of the coupled human-object system. They might modify system dynamics to better match a given task, possibly by adjusting their arm stiffness. This highlights the fundamental significance of resonance sensitivity in humans, which extends beyond object interactions to intentionally leveraging human body dynamics [2, 46, 47]. Even in social interactions, humans subconsciously entrain, e.g., to the step patterns of other humans while walking [48]. Humans may have evolved such a strong sensitivity to resonance because controlling resonant, and thus more predictable, motions could demand less mental and physical effort. Reducing energy needs is suggested as an essential factor for the evolution of human (loco)motion control strategies [49–51].

## Conclusion

For the first time, the presented research could provide data-based evidence that human resonance sensitivity extends to object interactions with nonlinear dynamics. The contribution of this research is twofold: first, we introduced methods that expand analysis options for human interactions with nonlinear systems. Second, the findings advance existing insights about human motor control strategies and their flexible adaptation to best suit a given task. Although interactions with more system variations should be investigated in the future, the experiments highlighted the fundamental importance of resonance for humans and their capabilities to excite it. Therefore, this research contributes to identifying underlying mechanisms driving human motion planning and control, especially during the excitation of periodic motions.

## Methods

### Ethics statement

The experimental human user study was approved by the Institutional Review Board of the German Aerospace Center. It included 20 right-handed participants (12 male, 8 female, 21–45 years). Prior to the experiment, the procedure and objectives of the study were explained to all participants, and written informed consent was obtained.

### Nonlinear modes of a compliant double pendulum

It is known that chaotic systems, like the well-known example of the double pendulum in gravity, exhibit quasi-periodic behaviors [52]. Some of these periodic orbits can be stabilized, e.g., through synchronization [53]. When paired with a dissipative system, chaotic controllers also demonstrate the ability to automatically entrain to the system through feedback resonance [47]. Extending this knowledge, recent advances showed that it is further possible to discover intrinsic periodic orbits of chaotic systems methodologically [28]. Using algebraic topology and differential geometry, the intrinsic dynamics of a nonlinear system can be analyzed, establishing three classes of periodic orbits: 1) toroidal orbits, 2) disk orbits, and 3) brake orbits. The latter category is especially interesting for the analysis of human interactions as they are an extension of normal modes known in linear systems [27, 28]. Research with simple linear systems has shown that humans are sensitive to such eigenmodes [21, 22] and might even apply the concept to control their own body [46, 47].

To introduce the concept of nonlinear normal modes (NNMs), we start from a simple linear case of a compliant pendulum system swinging horizontally, i.e., without gravitational influence. The pendulum link of length $l_1$ has the mass $m_1$ concentrated in the link's center and a rotational, linear spring with stiffness $k_1$ that exerts forces relative to the joint's equilibrium position $q_{eq} = 0$ (Fig 8A). The dynamics of this simple system can be described with

$$M(q)\ddot{q} + C(q, \dot{q})\dot{q} + K(q - q_{eq}) = 0 \ ,  \tag{1}$$

where $M(q)$ represents the mass matrix, $C(q, \dot{q})\dot{q}$ contains the Coriolis and centrifugal forces, and $K$ is the stiffness matrix. For the single pendulum, the equation simplifies to $C = 0$, $K = k_1$ and the mass matrix $M$ reduces to the scalar link inertia $J_1 = \frac{1}{3}m_1 l^2$ for the case that all the mass $m_1$ is concentrated at the center of the link of length $l$. The system will naturally oscillate at the constant (eigen-)frequency [23, 24]$\omega$:

$$\omega = \sqrt{\frac{k_1}{J_1}} \ . \tag{2}$$

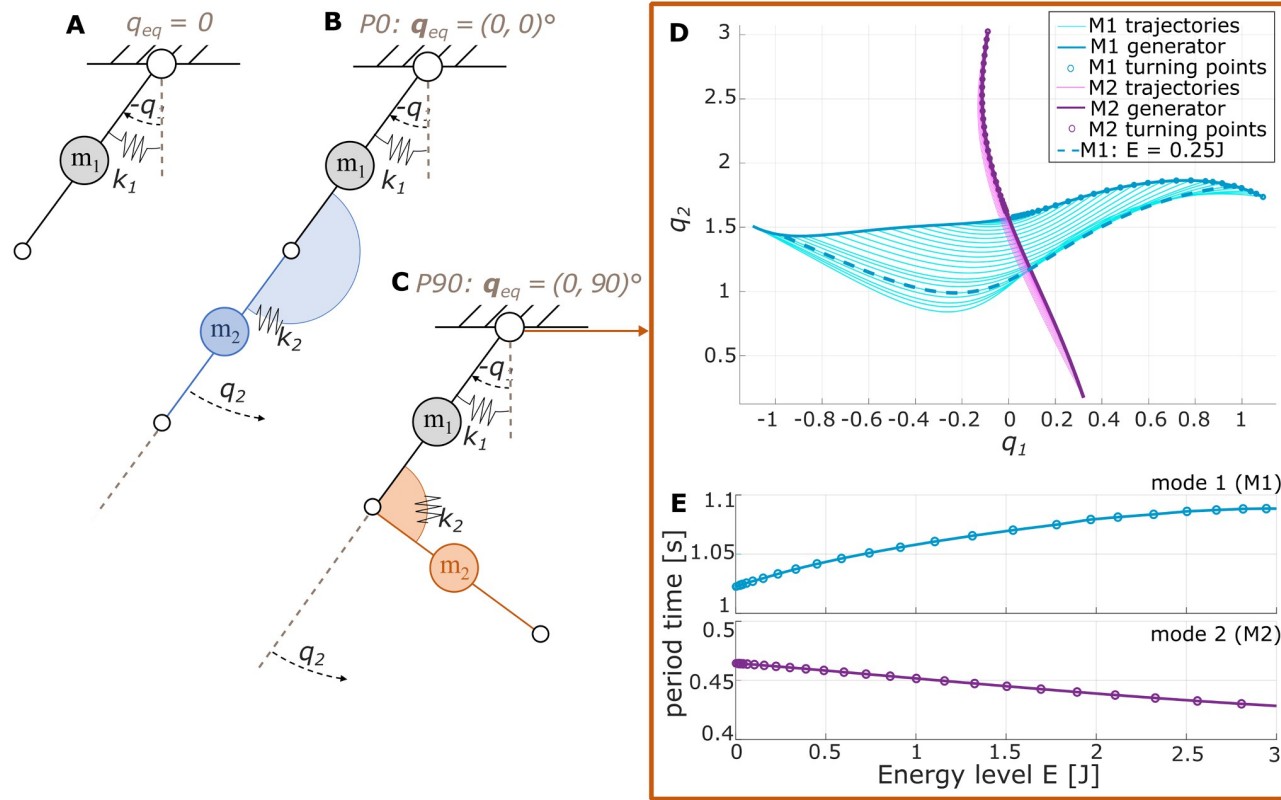

**Fig 8. Overview of pendulum kinematics and NNMs.** (A) Schematic simple pendulum with the link-centered mass $m_1$ and a rotational spring with stiffness $k_1$ in its origin. The spring equilibrium is set to the origin (dotted line) of the link angle $q_1$ ($q_{eq} = 0$). Adding a second link with mass $m_2$ through another spring of stiffness $k_2$ leads to a double pendulum, where the second link angle $q_2$ is defined relative to the first link. While the spring equilibrium at the first link always remains zero, different pendulum configurations can be defined by changing the equilibrium position of the second spring to either describe (B) a fully extended pendulum P0 with $q_{eq} = (0, 0)°$ or (C) a flexed pendulum P90 with $q_{eq_2} = (0, 90)°$. (D) Computing the nonlinear normal modes (NNMs) for the double pendulum configuration P90 reveals two modes, M1 and M2. A generator collects the turning points for the brake orbit oscillation on each energy level. The specific trajectory of the first mode for an energy level of $E = 2.5$ J is highlighted as an example. (E) Increasing energy changes the period times for both modes of the considered double pendulum system.

Connecting a second link of length $l_2$ and mass $m_2$ to the single pendulum through another spring of stiffness $k_2$ results in an elastic double pendulum, where $K = \text{diag}(k_1, k_2)$. Using Lagrangian formalism, the equations of motion for the flexible double pendulum can be derived in the form of (1).

Assuming the links' masses are concentrated in their centers, the mass matrix of the double pendulum reads

$$M(q) = \begin{bmatrix} J_1 + J_2 + \frac{1}{4}l^2(m_1 + 5m_2 + 4m_2\cos(q_2)) & J_2 + \frac{1}{4}l^2 m_2(1 + 2\cos(q_2)) \\ J_2 + \frac{1}{4}l^2 m_2(1 + 2\cos(q_2)) & \frac{1}{4}m_2 l^2 + J_2 \end{bmatrix}, \quad (3)$$

and the energy $E = T + V$ consisting of the kinetic energy $T$ and potential energy $V$ is

$$E(q, \dot{q}) = \underbrace{\frac{1}{2}\dot{q}^T M(q)\dot{q}}_{\text{kinetic energy}} + \underbrace{\frac{1}{2}(q - q_{\text{eq}})^T K(q - q_{\text{eq}})}_{\text{potential energy}}. \quad (4)$$

For the compliant case, changing the spring equilibrium leads to different configurations of the double pendulum with varying dynamics, e.g. $q_{\text{eq}} = (0, 0)$ shown in Fig 8b or $q_{\text{eq}} = (0, \pi/2)$ in Fig 8C. In contrast to linear normal modes, NNMs cannot be derived analytically from the equations of motion but need to be explored numerically [54]. To do so, the system is first linearized around the stable equilibrium $(q, \dot{q}) = (q_{\text{eq}}, 0)$:

$$0 = M(q_{\text{eq}})\ddot{\tilde{q}} + \frac{\partial^2 V(q_{\text{eq}})}{\partial q^2}\tilde{q}, \quad (5)$$

where $\tilde{q} = q - q_{\text{eq}}$. Decomposing this linearized system results in two oscillators with the frequency $\omega_1$ and $\omega_2$, respectively. The modes evolve along the respective eigenvectors $v_1$ and $v_2$, which follow the superposition principle for the linear case. For very small energies, these assumptions hold approximately true even in the nonlinear system, leading to periodic orbits with exactly two turning points. Between these points, the system performs rest-to-rest motions without deviation and without exhibiting chaotic behavior. This modal oscillation is denoted *brake orbit*. When increasing the energy level, the orbits of the NNM evolve nonlinearly, and the linearized assumptions no longer hold. Nevertheless, the two turning points can be slightly adjusted so that another periodic orbit emerges for every energy level. The adjustments needed can be made by applying numerical continuation methods. Starting from the linearized case for a minimal energy level, the energy is increased in small steps to find the next two turning points from which a stable brake orbit emerges. The collection of turning points that evolved from the *i-th* linearized mode leads to a continuous family of brake orbits. This family is considered as the *i-th nonlinear normal mode*. Summarizing the turning points in a function parametrized by the energy $E$ defines a *generator* $G_i(E)$ for every mode $i$. Initializing the system on $G_i(E)$ with zero velocity will always result in a periodic orbit that oscillates between the two found turning points. Since the brake orbit trajectories change for different energy levels, the period time $T$ varies with energy, such that

$$q(0) = G_i(E) \qquad \dot{q}(0) = 0$$
$$q(t) = q(t + T_i(E)) \qquad \dot{q}(t) = \dot{q}(t + T_i(E))$$

To apply the described procedure to arbitrary systems, our group has developed a *mode tool* [29, 30] that carries out the numerical continuation to obtain the NNM. To demonstrate the procedure, we specifically employ this tool to the double pendulum case for $q_{\text{eq}} = (0, 90)°$

**Table 4. Parameter values of experimental double pendulum configurations.** Only for a variation of Experiment 2, the second link mass $m_2$ was increased to 0.625 kg.

| first link | | second link | | unit |
|---|---|---|---|---|
| $l_1$ | 0.5 | $l_2$ | 0.5 | [m] |
| $m_1$ | 0.5 | $m_2$ ($\uparrow m_2$) | 0.25 (0.625) | [kg] |
| $k_1$ | 5 | $k_2$ | 3 | [N m rad$^{-1}$] |

(Fig 8C). Starting from the linearized assumptions in (5), two NNMs, *M1* and *M2*, can be developed for the system, which is visualized in Fig 8D in cyan and magenta, respectively. For both modes, a generator defines the turning points. Initializing the double pendulum from these points obtains the different brake orbits visualized by individual trajectories. As a specific example, we point out the brake orbit of the first mode for an energy level of $E = 2.5$ J with a dotted line. Fig 8E shows the development of the period times $T$ for the two modes over increasing energy levels.

Although the introduced concept of NNM is valid for the chaotic double pendulum under gravity conditions, the here introduced example case is arranged horizontally and incorporates springs in both joints. As such, the system is not intrinsically unstable if the joint stiffness is chosen high enough. While research suggests that humans can, to some extent, learn and anticipate the behavior of chaotic systems [1, 3, 55], this study focused on exploring the hypothesis that humans are sensitive to nonlinear normal modes and can purposefully exploit them. Thus, we chose to investigate the dynamically less complex version of the double pendulum swinging horizontally and with relatively stiff springs, which should be familiar to humans due to its similarity with the human arm. To not limit the investigation to one single system, both double pendulum configurations introduced in this section were considered (Fig 8B and 8C), corresponding to an extended and a partially flexed arm. In the following, the two configurations are denoted as *P0* and *P90*, respectively. The applied parameter values of the pendulum system were roughly based on the dimensions of a human arm (Table 4).

For both systems, the *mode tool* is applied to derive the NNM for one specific energy level $E = 2.5$ J that is arbitrarily chosen. Looking at the characteristic multipliers, i.e., the eigenvalues of the Poincaré return map, it can be identified that the first mode is more stable [54]. For this energy level, the computed period time was 1.29 s and 1.08 s for *P0* and *P90*, respectively. This translates to an eigenfrequency of the NNM of $f_{\text{res}(P0)} = 0.78$ Hz and $f_{\text{res}(P90)} = 0.93$ Hz. The resulting mode trajectories in joint space and Cartesian space are shown in Fig 9 for the *P0*

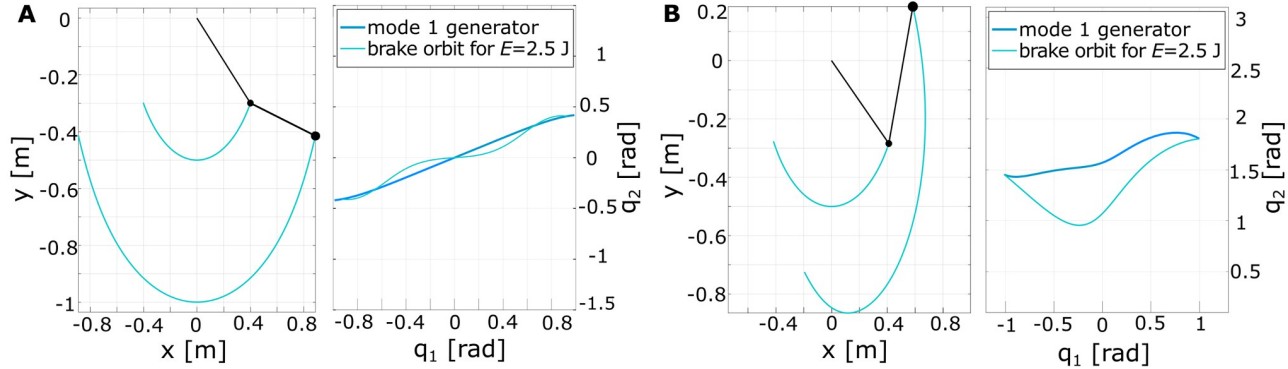

**Fig 9. Computed brake orbit trajectories for an energy level of 2.5 J.** The orbits are shown in Cartesian space (left) and joint space (right) for double pendulum configurations (A) *P0* and (B) *P90*. These brake orbits were used as a reference in the human user study.

(left) and *P90* (right) configuration. As visualized, the considered *P0* brake orbit is still relatively close to a linear solution. In contrast, the *P90* orbit is clearly nonlinear, such that the two configurations characterize varying nonlinearity.

### System implementation and characterization

For the human user study, the double pendulum systems were implemented in a virtual environment using Gazebo 11 with the parameters from Table 4. To drive the system, a visual motor link was added to the first pendulum link with a spring of stiffness $k_1$ (Fig 10, red link). The motor position $\theta$ thus defined the equilibrium position of the first pendulum link. To command this position in real-time, the simulation was coupled to a haptic 1-DOF joystick (Fig 10, left) that reflected the $k_1$- spring torque $\tau$ as feedback to the user. In this way, the user was given the impression of holding a real object and shaking it to excite oscillations mimicking a natural interaction. Friction was also added to both pendulum joints ($d_i = 0.02k_i$), such that a sustained control action of the human users was necessary to drive the system in a motion with a constant energy level. To define the energy level required in the experimental task, a ball geometry was added in two locations as targets that had to be reached. A custom Gazebo plugin changed the target ball color to visually indicate that the targets were correctly reached whenever a mesh collision with the second pendulum link was detected. No physical collisions between any of the bodies were defined, meaning that the first pendulum link could move through the motor link.

The control loop was running at 1 kHz and parameters were recorded with the same frequency. The recording included the angle positions and velocities of all joints as well as the reflected forces to the user and the displayed screen information.

To characterize the pendulum systems' response to different frequencies, a sweep was applied to the system through a simulated input to the motor link. Additionally, random motions were commanded to the motor link to outline the pendulum's reachable space, verifying that the systems do not automatically fall into the NNM without appropriate stabilization through control actions. In both cases, a sine wave with amplitude $A$ and varying frequency $\omega$

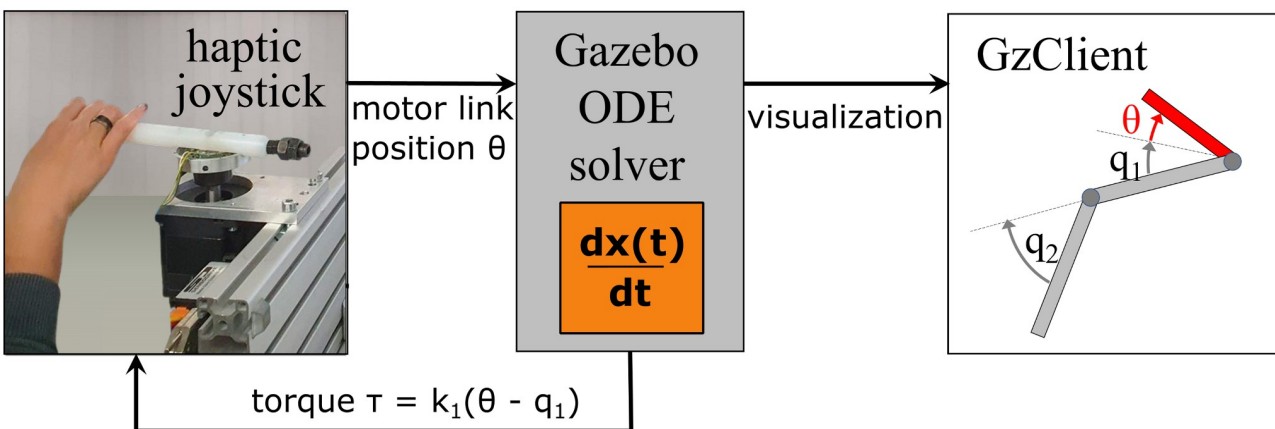

**Fig 10. Control scheme of the experimental setup.** Participants command the virtual double pendulum implemented in Gazebo through a joystick mapping 1:1 to the position $\theta$ of a motor link (red). Connected by a spring, this moves the first pendulum link ($q_1$), reflecting the spring forces $\tau$ to users as haptic feedback through the joystick.

**Table 5. Applied sine wave frequencies $\omega$ to characterize the $P_0$ and $P_{90}$ pendulum response with a sweep.** For each frequency, the sine amplitude $A$ of the handle motion was manually tuned to reach a set deflection of the first link ($q_1 \approx 1$ rad $\approx 60°$).

| $\omega$ | 1 | 2 | 3 | 4 | 5 | 6 | 7 | 8 | 9 | 10 | [rad s$^{-1}$] |
|---|---|---|---|---|---|---|---|---|---|---|---|
| $A(P_0)$ | 1.0 | 0.8 | 0.65 | 0.35 | 0.12 | 0.55 | 1.2 | 1.6 | 2.3 | 3.0 | [rad] |
| $A(P_{90})$ | 1.0 | 0.8 | 0.65 | 0.55 | 0.3 | 0.15 | 0.6 | 1.0 | 1.4 | 2.5 | [rad] |

was commanded as motor link position:

$$\theta = A \, \sin(\omega \, t) \, . \tag{6}$$

For the sweep, $\omega$ was varied from 1 to 10 rad s$^{-1}$ in steps of 1 rad s$^{-1}$ tuning $A$ manually for each frequency to reach the same angle deflection of the first pendulum link $q_1 \approx 1$ rad (Table 5). For the random control, the $A$ and $\omega$ were pseudo-randomly resampled every 0.1 s with $A \sim \mathcal{U}(1, 10)$ rad and $\omega \sim \mathcal{U}(0, 10)$ rad s$^{-1}$. In both experiments, the pendulum started from rest ($q_1 = 0$). The data was logged for 30 s per sweep frequency and for 60 s with the random control. The random scenario was repeated three times.

As expected, the sweep excited the largest oscillations of the link side compared to the motor link motion at values close to the eigenfrequencies $f_{res}$ of the respective pendulum configuration (Fig 11A). For *P0*, this was the case at 5 rad s$^{-1}$ ($f_{res(P0)} = 0.78$ Hz = 4.9 rad s$^{-1}$), while for *P90* it showed to be at 6 rad s$^{-1}$ ($f_{res(P90)} = 0.98$ Hz = 6.16 rad s$^{-1}$). At these frequencies, plotting the joint coordinates also shows the joint motions being closest to the expected nonlinear modes derived from the ideal conservative system. The generation of the random

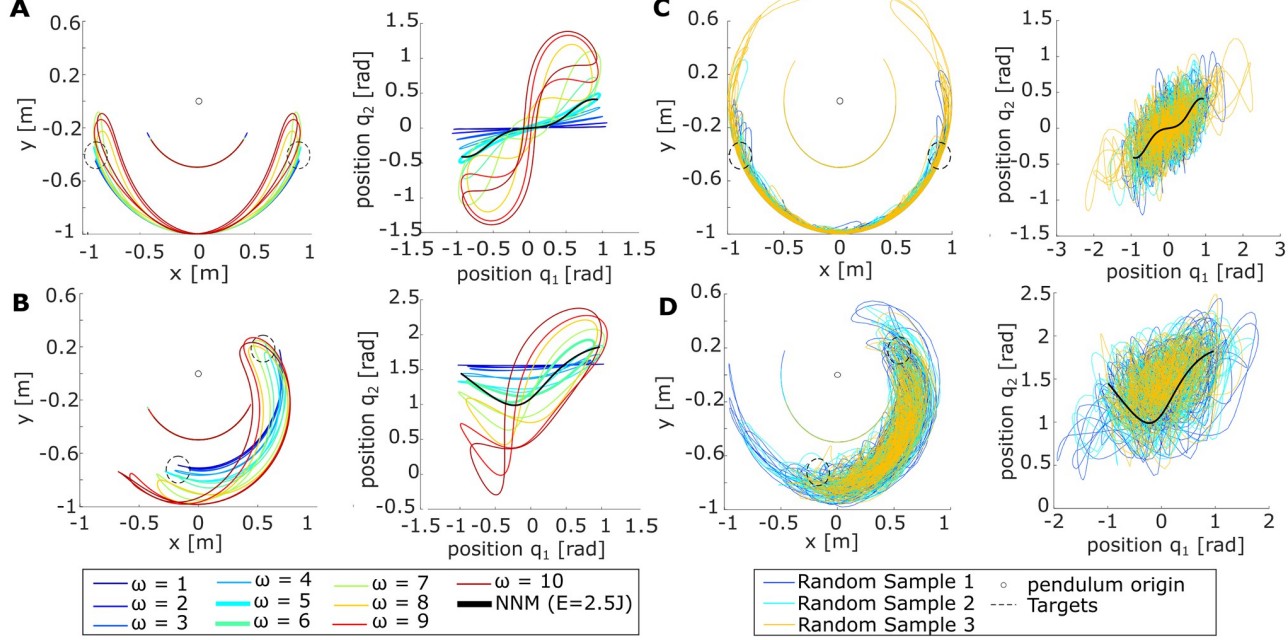

| — $\omega = 1$ | — $\omega = 4$ | — $\omega = 7$ | — $\omega = 10$ |
| — $\omega = 2$ | — $\omega = 5$ | — $\omega = 8$ | — NNM (E=2.5J) |
| — $\omega = 3$ | — $\omega = 6$ | — $\omega = 9$ | |

| — Random Sample 1 | ∘ pendulum origin |
| — Random Sample 2 | ---- Targets |
| — Random Sample 3 | |

**Fig 11. Characterization of pendulum responses and reachable space.** (A,B) To characterize the system responses to different input frequencies, the motor link position $\theta$ was commanded sine waves with different frequency values $\omega$ in simulation. (C,D) To outline the reachable space of the pendulum systems, the motor link was commanded a sine wave, where the amplitude and frequency were pseudo-randomly changed every 0.1 s. (A,C) show the *P0* and (B,D) the *P90* configuration. Respective plots on the left depict the system responses in Cartesian space, while the right plots show the corresponding joint space trajectories with the NNM in black ($E = 2.5$ J).

motions on the motor link visualizes that the reachable space of the pendulum system is quite large, which is especially apparent in the joint space plots (Fig 11B). Thus, the carried-out characterization measurements prove that the pendulum systems can be excited in various ways and do not automatically fall to their respective NNM.

## Experiment

**Task.**   The experimental task was designed as a game with the goal to collect hit points by alternately hitting two targets as often as possible within 40 s. This should motivate participants to move rhythmically and optimize the pendulum motions for speed to reach a high score. To validate the hypothesis that the human resonance sensitivity extends to nonlinear system dynamics, the experimental task was defined such that it had an energetic benefit to exploit the intrinsic pendulum dynamics, i.e., the NNM. Therefore, the targets were located on the turning points of the systems' NNMs (Fig 1C, blue/orange and Fig 9). When the second pendulum link entered the target region, a *hit* was visually indicated by a color change of the targets. The *hit* was rewarded with a point if the double pendulum did not swing through the target, i.e., did not leave the target on the opposite side of entering. In case of such an *overshoot*, next to the visual feedback, a beep sound was played as an error cue for the participant. This way, the participants were encouraged to maintain a constant energy level. The target $r_t$ determined the required accuracy to achieve the task. New hit detection was initialized whenever the second link crossed the equilibrium angle of the first link $q_{eq_1} = 0$, such that the two targets had to be hit alternately to collect points. Earned points through correctly hitting a target and the remaining time per trial were displayed to the participants through a Python-based GUI (Fig 1B).

**Variations.**   Different experiments were carried out to test the underlying human control strategies and their robustness. The task, instructions and trial time (40 s) always remained identical, but system dynamics or target size and locations were altered.

**Experiment 1.**   In the initial experiment, the *P0* and *P90* pendulum systems were parametrized as described in the modes section (Table 4). The targets were located on the turning points of the respective NNMs (Fig 1C, blue/orange and Fig 9) with a radius of $r_t = 0.1$ m to allow some leeway in the required hitting accuracy.

**Experiment 2.**   Experiment variations tested if and how participants altered their control strategy for *P0* and *P90* when task complexity increased. Two different task variations were investigated, each presented to half of the participants to avoid fatiguing. For the first group, the target radius was decreased to $r_t = 0.05$ m, such that the task difficulty in terms of accuracy was higher. For the second group, $r_t$ remained identical to Experiment 1 ($r_t = 0.1$ m), but the mass of the second pendulum link was changed to $m_2 = 0.625$ kg. The mass increase altered the systems' behavior and dynamics, which was noticeable to the participants due to slowed-down oscillations and increased feedback forces. Re-computed the NNM of the systems with altered mass with the *mode tool* showed that for the energy level of 2.3 J, the turning points still lay within the initial target radius $r_t$, such that the target positioning was not changed. Thus, participants could maintain an identical swing amplitude, which avoided visually biasing their control strategies. Participants were informed of the respective changes prior to the experiment run, but no change of the control strategies was suggested.

**Experiment 3.**   This experiment variation tested how participants adapted their control strategy when the intrinsic system motions were not aligned with the task, e.g., when the targets were not located on the turning points of the computed NNM. For this, a third pendulum configuration *P45* was implemented, where $\boldsymbol{q}_{eq} = (0, 45)°$, thus representing an intermediate configuration between the two initial configurations (Fig 12B). With the

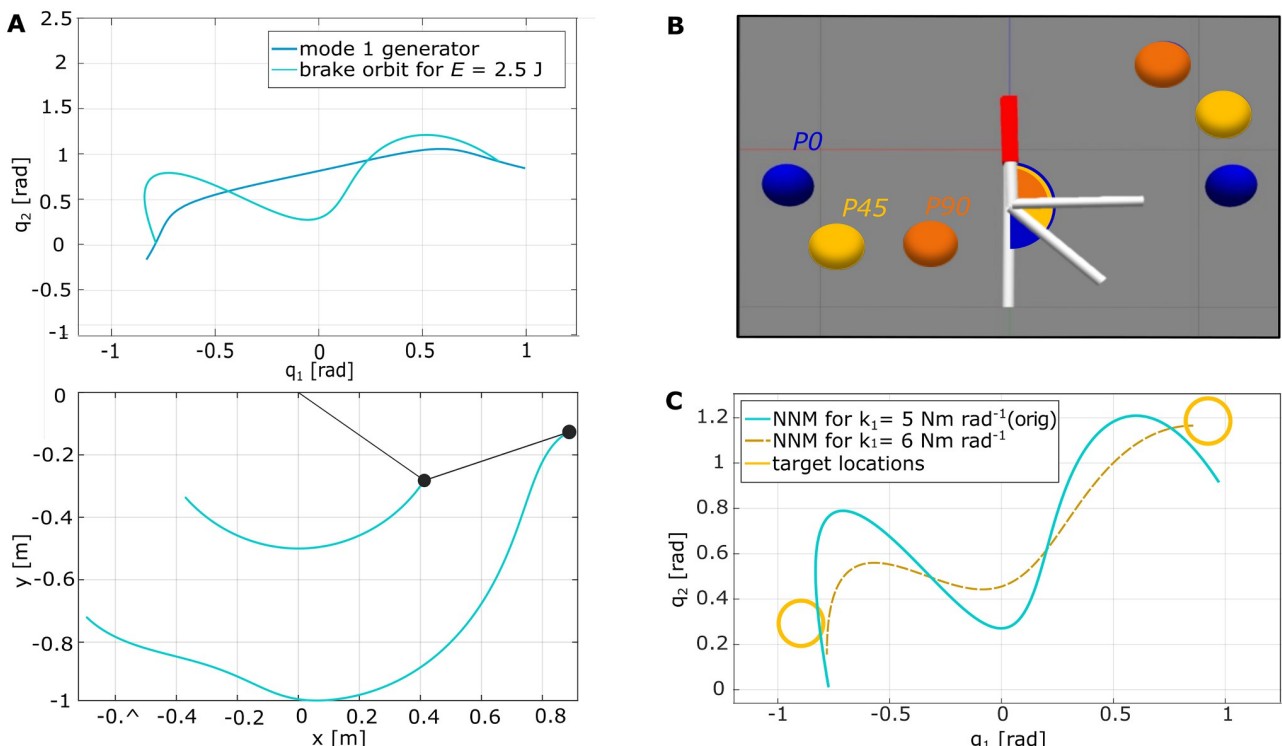

**Fig 12. Computed NNM and target placement for the *P45* configuration.** (A) Computed NNMs of *P45* with the initial parameters from Table 4 in position joint (top) and Cartesian space (bottom). (B) *P45* target locations (yellow) arranged between the targets of *P0* (blue) and *P90* (orange). (C) Comparison of the computed NNM for the original *P45* system (cyan) and the same pendulum with an adjusted first spring stiffness $k_1$ (yellow).

parameters from Table 4, the NNM of the conservative system was computed (Fig 12A). However, instead of locating the targets on the turning points, they were systematically aligned with the *P0* and *P90* targets (Figs 1C and 12b, yellow). By choosing a slight variation of the same system, instead of relocating the targets for the *P0* and *P90* configurations, biasing the users' control strategy for the previous two experiment variations was avoided. Instead, the three pendulum configurations *P0*, *P45* and *P90* were presented in randomly shuffled order within one experimental run leaving the participants unaware whether the targets were aligned with the NNM (Exp. 1) or not (Exp. 3). To help interpret the results of the user study, the spring stiffness $k_1$ between the first pendulum link and the motor link was altered empirically to see how the NNM develops. Solely, the first spring stiffness was altered since this parameter is the only one possibly affected when humans change their wrist stiffness. With $k_1 = 6$ N m rad$^{-1}$, the altered system showed an NNM that was within or at least very close to the *P45* target locations (Fig 12C, yellow).

**Experimental procedure.**   The participants were seated in front of the setup consisting of a screen and the haptic joystick (Fig 1A and 1B). They were free to orient themselves to hold the joystick handle most comfortably to them; only the elbow should be rested on the table to minimize fatigue. Initially, each participant had 5 m to familiarize themselves with the setup, the different pendulum configurations, and the task. No instruction was given on how to excite the system to hit the targets, but the participants could try their strategies during the familiarization period. When the experimental trials were started, the Gazebo simulation was initialized and displayed the initial position of one of the investigated double pendulum

configurations (*P0*, *P45*, *P90*). The simulation start enabled the force feedback on the haptic joystick and triggered the Python GUI, displaying the time countdown and current hit score. The participant had several seconds to start the task and tune into their control rhythm. Only then did the experimenter start the trial recording, triggering the 40 s-countdown and the hit counter. Thus, solely the human strategy to sustain the system oscillation was analyzed, neglecting the transient time. When the countdown reached zero, all windows automatically closed, and the force-reflection stopped. After the data recordings were saved, the experimenter manually triggered the next trial, making sure the participant was ready. The trials were repeated until all three pendulum configurations had appeared four times. The order of appearance was individually pseudo-randomized per participant. It was randomly selected whether a participant would first be presented with the initial pendulum settings (Table 4) used for *Experiment 1+3* or with the altered setup ($\uparrow m_2$ or $\downarrow r_t$) of *Experiment 2*. Nevertheless, overall, the experiments were selected in such a way that half the participants started with the initial settings and the other half with the respective variation setup. The complete user study took around 60 min.

## Baseline strategies for system excitation

Based on previous findings about human preferences and abilities in dynamic interactions with (non-)linear systems, possible control strategies that humans could use to drive rhythmic motions of the above-introduced double pendulum are hypothesized. Each considered strategy is defined as possible *baseline strategy (BL)* that will be applied to the system and later used as references to identify the underlying strategy of the participants in the user study.

**BL1 (Resonance).**   The first hypothesized control strategy is derived from observations indicating that humans are sensitive to resonance, which allows the reduction of large numbers of mechanical DOFs for easier movement coordination. In everyday actions, like jumping on a trampoline or bouncing a ball, humans effortlessly tune into system dynamics. Even the cardiovascular system seems to exhibit resonant properties and can be entrained to breathing patterns [56]. Sensitivity to intrinsic resonance appears also beneficial for object interactions as well as for learning the motor control of the human body itself [2, 25, 46, 47, 57]. Experiments with simple linearized systems validate this resonance sensitivity [21, 22] and also show that rhythmic system motions are much better predictable for humans when moving with the resonance frequency [7, 8]. As studies from Hogan and Sternad have suggested that humans seek predictability and low sensorimotor effort in dynamic interactions [16–18, 20, 41, 58], we hypothesize that humans intuitively excite the pendulum systems with its respective resonance frequencies. This would entail that the humans make effective use of the NNM of the systems. The associated baseline strategy *BL1* tested the investigated systems thus assumes a sine wave (6) with

$$\omega = 2\pi f \tag{7}$$

with *f* being the eigenfrequencies $f_{\mathrm{res}(P0)}$ and $f_{\mathrm{res}(P90)}$ for the *P0* and *P90* configuration, respectively, as computed for the conservative case. The amplitude was tuned empirically to $A = 0.12$ so that both targets were hit.

**BL2 (Position Control).**   When humans cannot properly determine the system dynamics with visual or haptic feedback to predict the system behavior, they have to rely more on the immediate position and force information they receive [59, 60]. Additionally, it is suggested that humans make use of motion constraints if possible to reduce their effort [31]. In these interactions, smoothness of hand and actuating forces are prioritized [32]. Thus, the alternative baseline strategy *BL2* aimed to model a slower control strategy. By synchronizing the

motion speed of the motor link handle with the first pendulum link, extensive spring deflections and rapid direction changes are avoided. In the specific regarded case, these motions entail low forces and smooth force curves. To capture this behavior, *BL2* again commands a sinusoidal motion to the motor link but with a much slower frequency ($f = \frac{1}{2}f_{\text{res}}$). In this way, the elasticity of the springs was not exploited, resembling more a (rigid) position control of the first link. The amplitude was once more tuned empirically to hit the targets ($A = 1.0$). It was expected that participants might apply this kind of strategy when first interacting with the system to scope the system dynamics and test their internal model. Additionally, it was suspected that this more reactive control might be needed when task complexity increases, e.g., in Experiment 2.

**BL3 (Bang-bang).**   It has been found that bang-bang control signals sufficiently model arm-reaching motions following a minimum acceleration with constraints principle [33]. Additionally, this control strategy matches the muscle activation patterns during such reaching tasks [34]. Similarly, in continuous periodic interactions, where humans were tasked to stabilize coordinated cyclic movements in a virtual standing compliant double pendulum, a bang-bang seemed to characterize the control strategy [35]. Adapting this control principle for robotic applications showed that it was also effective in driving highly efficient motions by exciting the system's intrinsic dynamics [36, 61, 62]. Thus, the third baseline strategy *BL3* modeled a bang-bang controller with a deadzone [63]. Based on the robotic controller implementation [36], a jump in the motor link was applied whenever a fixed torque threshold $\epsilon_\tau$ was crossed. In the experimental setup, the triggering torque $\tau$ was calculated based on the deflection of the spring between the motor link and the first pendulum link:

$$\tau = k_1(\theta - q_1) \ . \tag{8}$$

The constants for the threshold and the desired position command were tuned empirically to hit the targets leading to $\epsilon_\tau = 1$ and $\hat{\theta}_z = \pm 0.5$. Depending on the sign, the motor link coordinate $\theta$ was commanded to do a jump to a desired position $\theta_d = \hat{\theta}_z$ according to

$$\theta = \begin{cases} + \ \hat{\theta}_z & \text{if } \ \tau > \epsilon_\tau, \\[2mm] - \ \hat{\theta}_z & \text{if } \ \tau < - \ \epsilon_\tau, \\[2mm] 0 & \text{otherwise.} \end{cases} \tag{9}$$

### Analysis and metrics

**Data preparation.**   To quantify whether the participants applied the hypothesized control strategy that makes use of the system's NNM, the recorded data was post-processed with MATLAB2020b. For each double pendulum configuration (*P0*, *P45*, *P90*), the worst of the four presented trials per participant was identified based on the achieved number of hit points. This trial, usually the first one, was excluded from the data analysis to avoid bias through initial learning effects, loss of focus, or distraction. The transient time each participant needed to find their rhythm was not analyzed, but only the strategy for the alternate target hitting. To equalize the recorded data, each analyzed trial was cut such that all started on one target side and ended on the opposite. The three considered trials were then connected to produce one combined trial per pendulum configuration per participant. The trials were divided into separate periods, identified by the time points where the Cartesian velocity of the pendulum tip was maximal, i.e., when the pendulum crossed the zero-axis of $q_1 = 0$. In contrast, a Cartesian tip velocity of

zero indicated the pendulum's turning points. If this point lay within the target radius, a *hit* (= + 1) was counted for the participant. If the pendulum changed direction before reaching the target, it was classified as *undershoot*, while turning behind the target indicated an *overshoot*. Neither *undershoot* nor *overshoot* affected the hit score, and the assigned values were only used to differentiate which fault occurred.

**Comparative metrics.**   To compare the performance between participants and the overall performance compared to the investigated baseline strategies *BL1–3*, four metrics were defined. All analysis was carried out with MATLAB2020b.

**Hit Score.**   The participants' hit scores in the experiment were only of secondary interest but were evaluated to quantify the task success. Distinctions were made between successful *hits* leading to a point and errors, i.e., *undershoot* or *overshoot*, where the second pendulum link changed direction outside of the target radius. Comparing the participant scores to the theoretically achievable scores when oscillating with the predicted eigenfrequency of the NNMs indicated the humans' precision.

**Oscillation Frequency.**   The excited pendulum oscillation frequency is an important metric to validate that the human resonance sensitivity extends to nonlinear system dynamics. Thus, the oscillation frequency for each pendulum configuration was determined by finding the average period times between turning points for every participant. The overall oscillation frequency was then compared to the ideally expected values of the conservative system ($f_{\text{res}(P0)}$ =0.78 Hz, $f_{\text{res}(P90)}$ =0.93 Hz).

**Mode metric.**   The mode metric $\eta$ was introduced to quantify how well the intrinsic motions of the double pendulum systems were exploited. The idea was to compare the excited pendulum path with the path of the ideal NNM trajectory obtained for the conservative systems at an energy level of 2.5 J through a "distance" measure. Therefore, only periods where the targets were hit on both sides were included in the computation of this metric ensuring that participants had maintained the required energy level. This included the majority of the swing attempts as apparent from Tables 1–3. The position and velocity data of the pendulum in the considered periods were averaged per participant and regarded in joint space ($q_1$ vs $q_2$). Although a relation to the oscillation frequency could be suspected, the comparative mode metric should be independent of time to also allow comparison with *BL2–3*. For such time-independent comparisons, gait correspondences and movement similarities from motion capture data are often measured by the *Dynamic Time Warping* (DTW) principle [37, 38]. The DTW principle is similar to a nearest-neighbor comparison but ensures that all points along the curves are accounted for. Let's denote the recorded path data of the pendulum $q(i)$ while the path of the ideal NNM is summarized in $Q(i)$, where $i$ is discrete time. Both $D$-dimensional signals are interpolated to have the same length $N$ for comparison, and the turning point at the same target side is taken as the start point of the alignment. Instead of only taking the Euclidean distance for the monotonously increasing points of both position vectors, DTW finds for each time instance $i$ of $Q$ a corresponding time index $f(i)$ of $q$ ($f : \mathbb{N} \to \mathbb{N}$), such that

$$\bar{\eta}[f] = \sum_{i=1}^{N} \sqrt{\sum_{d=1}^{D} \Big( Q_d(i) - q_d(f(i)) \Big)\Big( Q_d(i) - q_d(f(i)) \Big)} \; . \tag{10}$$

is minimized. The mode metric used for comparison is the minimum value of $\bar{\eta}$ over all $f$ found by DTW, i.e., $\eta = \min(\bar{\eta})$. We used the MATLAB `dtw`-function for computation from the data. As the double pendulum's NNM is outlined by a manifold including first and second link positions and velocities $\boldsymbol{q} = [q_1, q_2, \dot{q}_1, \dot{q}_2]$, the distance quantified by $\eta$ compared all four quantities. Taking the ideal NNM trajectory as a reference, $\eta$ was determined for the individual participant data and for all baseline strategies.

**Handle Motion.**  In linear systems, resonant behavior is usually characterized by a phase lag of $0.5\pi$ relative to the period time $T$ between the input and output signal, along with an amplification of the output signal. Thus, we examined both these characteristics for the experimental data to gain insight into the individually applied control strategy per participant. The phase lag $\phi$ was computed between the motor link position $\theta$ as input and the first pendulum link coordinate $q_1$. This calculation was performed using cross-correlation with the MATLAB `xcorr`-function, from which the sample lag in seconds can be obtained. As participants and baseline strategies varied in period time $T$, the lag times were expressed relative to $T = 2\pi$ to allow comparability. To calculate the deflection ratio $\rho$ between the input and output signal, the absolute peaks $\max(\theta)$ and $\max(q_1)$ per period were identified, and their quotient was determined.

**Statistics.**  To investigate the human resonance sensitivity for nonlinear systems, all comparative metrics presented in the Tables 1, 2 and 3 were tested statistically. Experiments 1 and 3 were carried out with all participants ($n = 20$), while the two variations in Experiment 2 only included half participants ($n = 10$). Each participant sample consisted of the appended data from the three best trials as described above. All statistical tests were carried out with the dedicated functions of MATLAB2020b. Initially, we verified the normal distribution of the participant samples for all metrics with the one-sample Kolmogorov-Smirnov test ($\alpha = 5\%$). For all following statistical analysis, the difference between the individual participant values per metric and the respective values obtained for the baseline strategies were calculated. This was necessary since the baseline strategies were deterministic controllers instead of samples, such it was not possible to treat them as different sample pools. Solely the differences, denoted $\Delta BL1$–$3$, between the participant metrics to each of the baseline strategies could be analyzed statistically.

To first investigate whether the different baseline strategies were different from each other, a one-way ANOVA was applied (`anova1`) to the data of Experiment 1, followed by a post-hoc pairwise comparison using Tukey correction (`multcompare`). All results showed very strong significance verifying the assumption that the tested baselines were fundamentally different. The corresponding statistics are reported in Tables A and B in the S1 Table.

To test the similarity between the participants' applied control approach and each of the baseline strategies, the differences values $\Delta BL1$–$3$ for all metrics were statistically compared with a two-sided one-sample t-test (`ttest`). If the participant control was similar to one of the strategies characterized by $BL1$–$3$, the mean difference was expected to not differ from zero. Specifically, the oscillation frequency was of interest, since our hypothesis ($H_1$) stated that humans would intuitively excite resonance in the pendulum systems. Comparing the differences of the mode metric values could further indicate whether the applied participant control lead to pendulum motions that were similar to the ones excited by one of the baseline strategies. Thus, the reporting of the statistics in the results mainly focuses on the comparison of these two metrics, but all statistics are presented in the Tables C, D and E of the S1 Table. Since the participant data was compared to $BL1$–$3$ individually, the initial $\alpha$-level of 5% had to be adapted to correct for the multiple comparison of one sample. Thus, the significance level was lowered according to Bonferroni for all metric-comparisons such that

$$\alpha = \frac{\alpha_{5\%}}{3_{BL}} = 1.7\% \ . \tag{11}$$

Additionally, we examined if the applied phase lag value $\phi$ per participant had a systematic influence on other metrics, namely the individual hit scores and mode metric values. Using the Pearson correlation (`corrcoef`), it was investigated whether the relation with the individual respective hits or $\eta$-values showed any significance.

## Supporting information

**S1 Table. Supplementary statistical results.**
(PDF)

## Author Contributions

**Conceptualization:** Annika Schmidt, Marion Forano, Arne Sachtler, Davide Calzolari, David W. Franklin, Alin Albu-Schäffer.

**Data curation:** Annika Schmidt.

**Formal analysis:** Annika Schmidt, Arne Sachtler, Davide Calzolari, Bernhard M. Weber, David W. Franklin, Alin Albu-Schäffer.

**Funding acquisition:** Annika Schmidt, David W. Franklin, Alin Albu-Schäffer.

**Investigation:** Annika Schmidt, Marion Forano.

**Methodology:** Annika Schmidt, Marion Forano, Arne Sachtler, Davide Calzolari, Bernhard M. Weber, David W. Franklin, Alin Albu-Schäffer.

**Software:** Annika Schmidt.

**Supervision:** David W. Franklin, Alin Albu-Schäffer.

**Validation:** Annika Schmidt.

**Writing – original draft:** Annika Schmidt.

**Writing – review & editing:** Annika Schmidt, Marion Forano, Arne Sachtler, Davide Calzolari, Bernhard M. Weber, David W. Franklin, Alin Albu-Schäffer.

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
