## [Decision Letter · Decision Letter 0]

12 Nov 2023

Dear Ms. Schmidt,

Thank you very much for submitting your manuscript "Finding the rhythm: Humans exploit nonlinear intrinsic dynamics of compliant systems in periodic interaction tasks" for consideration at PLOS Computational Biology.

As with all papers reviewed by the journal, your manuscript was reviewed by members of the editorial board and by several independent reviewers. In light of the reviews (below this email), we would like to invite the resubmission of a significantly-revised version that takes into account the reviewers' comments.

I apologize for the lengthy review process but it took many attempts to secure two willing reviewers and the there was some delay in the review process due to the complexity of the manuscript. Both reviewers expressed positive views of the paper but also require major revisions. Thank you for your patience.

We cannot make any decision about publication until we have seen the revised manuscript and your response to the reviewers' comments. Your revised manuscript is also likely to be sent to reviewers for further evaluation.

Sincerely,

Bard Ermentrout

Academic Editor

PLOS Computational Biology

Daniele Marinazzo

Section Editor

PLOS Computational Biology

I apologize for the lengthy review process but it took many attempts to secure two willing reviewers and the there was some delay in the review process due to the complexity of the manuscript. Both reviewers expressed positive views of the paper but also require major revisions. Thank you for your patience.

Reviewer's Responses to Questions

**Comments to the Authors:**

Reviewer #1: The manuscript “Finding the rhythm: Humans exploit nonlinear intrinsic dynamics of compliant systems in periodic interaction tasks” reports experimental evidence that human participants find and exploit efficient and periodic trajectories in the control of a multi-link dynamic system that is prone to chaotic behavior. This is an interesting study with a rich and novel behavioral, applied mathematical, and computational content. The main finding is intuitive but not trivial, and as such would be interesting for a diverse audience.

I have three substantial comments, but I believe these can be addressed. 1) This is much more of a behavioral study than a computational study and as such would be more suitable for the main PLoS than PLoS Comp Bio. It is evident that the study is computationally rich, it’s just that the latter part is hidden. Please consider if you could put more of the technical part in the text. 2) I think the manuscript can benefit from re-writing certain parts. It must be possible to convey the necessary information with fewer words. If this is achieved, then it would leave space for specifying the mathematical and computational aspects. 3) My greatest concern is with the form of the evidence which is not as strong as it could be.

The authors state that there is little literature on their topic, "... data-based evidence proving that humans indeed match system dynamics has been lacking." I believe there is a little more than what’s listed. Please include literature on feedback resonance and entraining of chaotic dynamics in human motor control. The latter involves a popular problem in the control of chaotic systems, namely stabilizing unstable periodic orbits embedded in a chaotic attractor, and this seems to be similar to the ideas in the present manuscript.

It is not obvious how the main hypothesis was tested. In fact, what was the main hypothesis and its null hypothesis? Only after I got to the results did I find that the evidence is in the form of comparing alternative control strategies (BS1, 2, 3). Furthermore, these comparisons do not appear to have been done statistically but just by eyeballing the means (“… is … visually apparent”, l. 285-7).

Even so, the comparisons are not convincing. For example, it looks like the human is closest to BS1, not to the theoretical mode (Fig 4). This is consistent with Table 1 where it appears that the human participants and BS1 were about equally far from the nonlinear mode reference, implying that participants obeyed control strategy #1. (Is it correct that the mode metric would be 0 if the participant's trajectory was exactly overlapping with the nonlinear mode?)

As I see it, the evidence is confirmatory and not very strong. If I understand correctly, the nonlinear mode is also the most energy-efficient trajectory, so it is not surprising that participants converged to it. The main empirical finding reported is that participants could exploit efficient control of nonlinear systems. But this is already stated as a matter of fact at the beginning, so the evidence merely confirms what we already know. The fact that participants found an efficient performance trajectory doesn't tell you much about how they executed it. This is not a bad flaw for a modeling study, it's just important to be explicit about it.

With relevance to interpreting the significance and results of the study, how hard was it to discover the good trajectories? It appears like this is a learning task, or at least a mix of existing motor skill for control of elastic objects and learning the specific constraints of this object. Were the participants experts in this task from previous studies?

I think the writing style can be changed so that it’s easier to get to the essential information. I didn’t understand the task, the nonlinear modes, and the statistical analysis, or lack thereof, until I was on page 9. The Results section is more of a narrative that mixes methods and observations.

The task, as described early in the paper, is not trivially intuitive for naive readers. Figure 1 helps with illustrating the task, but it could be improved. Could you show the axis of rotation of the joystick? How quickly does the pendulum travel? Is it possible to visualize the space of possible trajectories, without cluttering the screen? What would these modes correspond to in an intuitive example with an object used in daily life?

The labels "init, size, mass" in Table 1 are confusing. Better label as Exp 1, 2, .. or something like that, and explain in the captions that the runs correspond to manipulations of size and mass.

There are also small mistakes that need careful proof-reading for grammar and style. Some examples:

"... show that humans seemingly effortless[ly] support systems in their intrinsically preferred motions."

"All participants intuitively exploited the elasticity of the system by choosing a holding strategy of the motor link and only compensat[ing] for energy losses with small motions."

"... the nonlinear dynamics of [a] system and intuitively ..."

Reviewer #2: Summary

An experimental paradigm was developed to study whether humans exploit the passive dynamics of complex nonlinear systems to facilitate task execution. Human participants were asked to use a double pendulum in a virtual environment via a joystick to alternately “hit” two targets with the tip of the pendulum. The joystick provided force feedback. Three separate experiments were conducted:

Experiment 1: The human control strategy was compared against three baseline strategies: resonance, position control, and bang-bang control.

Experiment 2: The task difficulty was varied by reducing target size and increasing pendulum mass to study whether humans adjusted their control strategy.

Experiment 3: The system dynamics were altered so that the eigenmode of the system did not overlap with the targets. Thus, participants could not solely rely on the passive dynamics of the system to successfully execute the task.

Small levels of friction were added to make sure that participants continued to inject energy into the system to maintain its oscillation. The amplitude of the target behavior was experimentally maintained by instructing subjects to not overshoot the targets. The evaluation of human performance relied on a new computational method that could determine the eigenmode of the nonlinear system, although only in a conservative system.

Experiment 1 showed that humans employed control strategies that closely matched the resonance baseline strategy. This supported the hypothesis that humans exploited the eigenmodes of the system.

Experiment 2 showed that humans significantly altered their control strategy by reducing the oscillation frequency of the system when faced with increased task difficulty (smaller target size or increased mass). However, the authors argue that this change in frequency was small, and thus the overall control strategy had not changed.

Experiment 3 showed that the participants were able to execute the task with high success rate despite the targets not being located on the trajectory of the system’s eigenmode. This shows that humans only exploited the passive dynamics of the system when it indeed matched the task demands.

Major Comments

This paper is noteworthy as it addresses a new problem: how humans interact with and control a nonlinear underactuated system. The study offers a quantitative demonstration that humans exploit the passive dynamics of nonlinear, underactuated systems. That humans are sensitive to resonance or eigenmodes has been demonstrated in a few studies using simple mass-spring or pendulum systems. However, the eigenmodes of nonlinear systems are much less straightforward, both for the scientist to determine or for the human to sense. Hence, this is an important step forward, both computationally and experimentally. The experiments were thorough and the progression of the three experiments were well justified and alternative hypotheses were tested. As such, this paper is a first demonstration that the notion of ‘resonance tuning’ in human interactions with physical systems holds for complex underactuated systems that can display chaotic dynamics.

While the experiments are well executed and the results interesting, the manuscript needs a lot more work to be ready for publication. One major drawback is the writing that requires significant revision. Aside from numerous grammatical errors and odd sentence constructions, there is considerable redundancy in content and convoluted sentences sometimes make it hard to understand the meaning. As a broad yardstick, the text can be reduced by at least 30%. I would highly recommend that a native speaker goes over the writing.

The researchers cite four of their previous papers [17, 18, 20, 21] that were used to obtain the eigenfrequency and eigenmode of the double pendulum. As the entire study is anchored in this analysis and the analysis is new to the human neuroscience audience, a short explanation of this new approach should be included in an appendix.

Another major limitation is that the study lacks statistical evaluation of the hypotheses. For example, the results in experiment 1 should statistically compared with the three baseline strategies, such that the conclusion that the resonance strategy best describes human performance is statistically supported. A qualitative description of the mode metric values is not sufficient.

As a follow-up to this result: the alternative strategies are neither sufficiently motivated nor evaluated. Why did the authors choose these 2 alternative baselines control strategies? Is bang-bang a realistic strategy and something that humans could even reproduce? In addition, does the resonance baseline strategy align with energy minimization? Probably yes, but discuss the closeness to energy minimization. The performance of the alternative baseline strategies is not provided. How many hits are achieved when adopting BS1, BS2 or BS3? Are they similarly efficient in terms of performance (and given the time constraint)? If BS1 is more performance-efficient, it should be mentioned and discussed: humans may first choose performance efficiency before caring about energy.

The variable of the phase lag is not sufficiently motivated and explained in the methods. Similarly, the mode metric, which is in principle plausible, but the details evade me in the description.

Another specific question: Fig 4a (left) shows a significant difference between the theoretical mode and the trajectory for BS2. The only intersecting region lies close to the origin (q1=0, q2=0). Thus, the target locations that can be easily reached using BS1 (because the locations were designed to lie on BS1) would not be reachable with BS2. Can the authors clarify?

The results of experiment 3 are vital to demonstrate that humans can achieve the task in different ways if the task goals cannot be achieved with the eigenmode. But what are humans do in this case? While reverse-engineering the pendulum that humans effectively implemented, how did they do it? This gets me to a deeper point: the eigenmode analysis is for the conservative system, which is evidently an approximation. Not only does actual pendulum contain damping, the human hand and arm also contribute to the system’s properties, adding stiffness and damping. Hence, should we conclude that in experiment 3 the human changes its impedance to overall achieve this different behavior of the double-pendulum? The implications of these results are not sufficiently clear.

Another potentially interesting aspect that may strengthen the data is to examine whether participants are converging on a solution with practice.

The discussion needs to elevate the results to a higher level why and how humans interact with nonlinear dynamic systems.

Line-by-line Comments

Abstract: grammar - “humans seemingly effortless support” -> effortlessly.

Line 22-23: No direct connection between the text and the study in ref [8].

Line 22-23: Says “humans might excite the simple intrinsic modes of the system to hit a target”. The concept of modes has not been introduced, making this statement irrelevant and confusing.

Line 36-37: The relevance of [12, 13] to the prior text is unclear. Must briefly discuss the cited work to give better content.

Line 127: What creates the different energy levels? They introduce this later, but I believe this caused lots of confusion when interpreting figure 2 and the definitions of eigenmanifold, eigenmode, etc.

Line 170: Why was 2.5J energy level chosen? Why not a different energy level? Was it the total physical energy of the conservative system? A clearer description of what the energy level physically means would be informative.

Ln 195-196: What is different from the initial pilot experiments that led participants to not use a position control strategy? Was the same observation made in the reported experiments during the early trials?

Line 202: Insufficient justification for use of bang-bang control in this study. Are there existing studies that suggest humans use such control strategy?

Line 266-277: Extremely dense paragraph. Hard to follow.

Line 367: Provide information what statistical test was conducted that obtained p-values. See comment about absent statistics.

Line 417: when reverse-engineering the double pendulum, it is not clear whether there was more than one solution or whether the mass and spring constant were explicitly chosen for any particular reason. Clarification would be helpful.

Ln 444-448: The reported behavior seems to indicate an eigenmode: small input leading to a large output and the oscillatory behavior traverses the same trajectory repeatedly in the q1-q2 space. Can the authors clarify if they would call this an eigenmode?

Line 505: This study is not the first one to show how humans can intuitively interact with nonlinear dynamic systems (cf Turvey’s studies, that should be cited as well). But this study digs deeper into this interaction by investigating interaction with more complex systems (double pendulum).

Ln 517 – 524: It is unclear how the discussion point (1) is relevant to the argument made about “why humans choose to exploit the inherent dynamics” of the mechanical system. Are the authors trying to convey that participants excite the system at the eigenfrequency (indicating exploiting dynamics) since the accuracy requirement was not high enough. It counters the statement from experiment 2 that participants still exploited the intrinsic dynamics of the system when the accuracy requirements were increased. Maybe the authors can rephrase and clarify this paragraph.

Line 573 to 577: It is first claimed that the strategy was not altered, and the following sentence starts by ‘accordingly, the participants might likely alter their strategy […]’ it seems contradictory. Please clarify.

Line 585: what does this head mean?

Line 595 to 597: this passage starts by ‘the experiment data suggests that a phase lag around 6pi might be the best choice’, but the next sentence continues with ‘the present data is inconclusive’. Please clarify. It is unclear how the phase lag can be independently chosen by the participants. Should the phase lag not naturally arise with the type of excitation i.e., depend on the nonlinear phase response. For instance, with a linear system, each excitation frequency is also associated with a phase lag. Can the authors clarify if it is possible to dissociate the relationship between excitation frequency and phase lag for a nonlinear system?

Comments on Figures

Figure 1: The perspective on the hand in relation to the monitor display of the double pendulum makes it hard to get a sense of the task.

Figure 2, panel (d): What is plotted on the y-axis? I understand that this compares with panel (e).

Maybe adding better labels for the axes and colored lines within the figure and not buried in the wall of text that is the caption.

Figure 3: the gray straight lines in panels (a) and (b) are not explained. I think looking at panels (c) and (d) first before (a) and (b) would be easier to interpret. Switching the order of the panels would be easier to interpret.

Figure 6: the pendulum symbol P0 and P90 is different – legacy?

Figure 10: the label says ‘adjusted mode’ - please explain.

Figure 9, panel (a): the gray line as in Figure 3 are not explained. Also, the order of the panel labeling is a little confusing.

**Have the authors made all data and (if applicable) computational code underlying the findings in their manuscript fully available?**

Reviewer #1: Yes

Reviewer #2: **No: **Interesting study, but the presentation needs a lot more work.

PLOS authors have the option to publish the peer review history of their article (what does this mean?). If published, this will include your full peer review and any attached files.

Reviewer #1: **Yes: **Dobromir Dotov

Reviewer #2: No
---

## [Decision Letter · Decision Letter 1]

7 Mar 2024

Dear Ms. Schmidt,

Thank you very much for submitting your manuscript "Finding the rhythm: Humans exploit nonlinear intrinsic dynamics of compliant systems in periodic interaction tasks" for consideration at PLOS Computational Biology.

As with all papers reviewed by the journal, your manuscript was reviewed by members of the editorial board and by several independent reviewers. In light of the reviews (below this email), we would like to invite the resubmission of a significantly-revised version that takes into account the reviewers' comments.

Please address the major comments of Reviewer 2. Based on your reply, I will decide whether it needs to be sent out again.

We cannot make any decision about publication until we have seen the revised manuscript and your response to the reviewers' comments. Your revised manuscript is also likely to be sent to reviewers for further evaluation.

Sincerely,

Bard Ermentrout

Academic Editor

PLOS Computational Biology

Daniele Marinazzo

Section Editor

PLOS Computational Biology

Please address the major comments of Reviewer 2. Based on your reply, I will decide whether it needs to be sent out again.

Reviewer's Responses to Questions

**Comments to the Authors:**

Reviewer #1: The manuscript reads nicely. I think the abstract could still benefit from some work. I asked my partner, a scientist in a different field, to read the abstract and the response was, "They lost me at the first sentence".

Statistical analysis:

l 72-74 and below: Please verify that you've indicated the NULL and ALTERNATIVE hypotheses right. This looks more like the alternative hypothesis to me. If I understand it, BS2 and BS3 play the role of null hypotheses. You're trying to reject these to confirm the alternative model, BS1, which is the one that involves sensitivity to the resonant frequency. If this is correct, then a simple statistical test would be to show that eta_BS1 is significantly lower than eta_BS2 and eta_BS3. This is contingent, however, on showing that the simulations of each of the three baseline strategies used the best parameter estimates. Is this kind of optimization of simulations what is being referred to as "tuned empirically" in the "Baseline strategies ..." section, l. 706+?

More sophisticated statistical approaches are possible too. For example, fitting separately each of the BS1,2,3 models to the empirical data using MLE makes it possible to use the Bayesian information criterion to compare the three models, taking number of parameters (model complexity) into consideration as well. This will simply give you the best model. I don't know if this approach is feasible here but it would make for even stronger evidence.

A minor comment: Trampoline jumping and stick balancing are not exactly common everyday activities. Would bouncing a ball qualify as a valid example that is more relevant as a task of daily living?

Reviewer #2: See complete longer review in attachment. Comments on statistics not included here.

The authors provided an extensive revision of their manuscript in response to our comments. The re-writing of the manuscript has significantly improved readability. The authors have further strengthened some of their arguments by adding references and clarifying metrics, figures, and analyses. While this revised manuscript is a significant step forward, there is still a long list of comments and suggestions to further improve the manuscript. While the list is long, the study is scientifically valuable and, with more work, will hopefully become more rigorous in its presentation.

Major Comments

1) While the writing has improved, many passages, especially in the discussion, are still quite wordy and ‘unconstrained’. Tightening the language would improve readability. Some choices of words and phrases still indicate that a native speaker should go over the text.

2) The results are inconsistently reported with seemingly select statistics and the results of Experiment 2 are only reported visually. The human data obtained in the three experiments should be consistently compared with the three hypothesized strategies. The authors have introduced three metrics: task success, mode metric and phase lag. For each experiment, each of these metrics can be compared with the three hypothesized strategies. Coupled with the result that for example the mode metric was lowest for BS1, it can be inferred that participant trajectories were closest to the ideal NNM.

3) The experimental task is still unclear and difficult to understand, especially as different aspects of it are shown in three different figures. While the task variations are clear (reduced target size, etc.), the way the task is carried out by a human remains hard to visualize. For example, did the subject move their arm laterally to move the apparatus, or did they rotate the wrist, and in what direction? Provide more description on exactly how the user manipulated the apparatus. Combining at least the first and second figure will help.

4) The discussion of the control strategies is sometimes contradictory. The frequencies by themselves are not sufficient indicators of whether participants perform feedback (reactive) or feedforward control (prospective) as the authors also argue that participants could be modulating their arm stiffness to shift the NNM mode to lower frequencies. The more relevant difference between BS1 and BS2 is that BS1 requires low forces (low input, thus resonance and using NNM) and BS2 requires greater forces (large input, not exploiting resonance and not using NNM). It may be beneficial to have a different name for BS2. In Experiment 2, the authors argue that participants shifted the oscillation frequency farther away from BS1 to become more accurate. Does this not negate the hypothesis of the authors? If participants want to be more accurate, they should more accurately match their oscillation frequency to the “resonant” frequency of the system.

5) The discussion section talks about aspects of the data that were not reported in the results. The sinusoidal shape on the human trajectory, in line with the BS1 strategy and the initial explorative movements that were not shown in the results. These data should either be shown or eliminate reference to these data aspects in the discussion.

6) The target hit score seems somewhat biased. It is argued that overswing would alter the energy level and therefore would confound the comparison with the NNM. Do undershoots or missed targets not lead to a lower energy mode? If so, should they not also count as -1 as the overswings? A score report that splits between hits and misses, both over and undershoot may help clarify this. A different count of the hits may also reveal different patterns in the correlations with the phase lag.

7) The two mode metrics seem to obtain different results when compared with the NNMs. This is confusing. It appears that the mode metric of the full state space is the more complete one, hence the one that should be reported. While some differences may be less favorable for the hypothesized strategy, I assume the new suggested statistical comparison will render new and likely supportive results for BS1.

8) Overall, the concept of energy levels must be discussed in more detail. It is still unclear to me how different energy levels are achieved/excited. Is it by adjusting the initial angles of the pendulum, or by introducing a non-zero velocity as initial condition or?

9) The simulations for BS2 and BS3 should be explained more prominently as they serve as reference or alternative hypothesis throughout the paper.

Detailed Comments (following the text):

Introduction

Line 5 onwards: I appreciate the authors to better situate their work in existent understanding of human motor control. As I am keenly interested in this area of study, I probed into the cited papers. Unfortunately, I found some of the new references non-optimal. For example, amongst the references for predicting and following chaotic behavior, reference 2 is a non-refereed 4-page extended abstract. A similar extended abstract is cited later. For a journal like PLoS Comp Biology, it may be better to use peer-reviewed papers, especially in the introduction. The paper by Goodman et al (ref 9) seems more to the point here. Ref 5 is not really on resonance, may be better Kugler and Turvey (1987) and Raftery, Cusumano, & Sternad (2008). I also do not think that reference 7 is a good reference for resonance and dynamic stability.

Line 16: This point was also made by Maurice, Hogan, Sternad (2018).

Line 20: Reference 20 is again an unrefereed extended abstract, in the current context, ‘prospective control’ appears misleading as it was only an experimental manipulation, not a reliable assessment or simulation of prospective control. See also Sharif Razavian, Sadeghi, et al (2023) for this point.

Line 26: See also Nasseroleslami, Hasson, Sternad (2014) for the comparison between predictability, stability and smoothness.

Line 67, Figure 1: I fail to see the blue, orange and yellow configuration in the figure? Only the targets are colored and the pendulum links are white. Why not simply showing three pendulums separately? Also make the text follow the figure: first explain the left panel. Also, is the hand orientation really matched with the hand figure? Then, the hand would overlap or ‘collide’ with the double pendulum. Would it not make more sense to have the hand coming from the right to hold the red handle? Unnecessary: the hand looks as if it is manipulated by the left hand. Are the indicator panels for the remaining time and target hit score really as large? Figure 12 may be combined with Figure 1 to not distribute different illustrations of the experiment throughout the paper.

Line 75: I would not call the strategies ‘baseline strategies’, later abbreviated to BS. First, the acronym BS has negative connotations in English and American slang. I suggest to call them hypothesized strategy for the resonance tuning one, and alternative strategies for the other two. Or just short names. I also have reservations about strategy 2 as explained in the general comment above.

Line 81: The two metrics are not a contrast, delete ‘on one hand’ and ‘other hand’.

Results

Line 105: It may help readability to summarize the experimental task for the subjects, including the duration and scoring achieved. Also, state the exact conditions on Exp 1, 2, 3. It would also help to have a brief summary of the simulated strategies, BS1, 2, 3.

Line 112: It is typically useful to first state how participants achieved the task. In this case, the explicit instruction was to hit targets. Hence reporting target scores would be good. May be also list overswings and hits separately so that the reader gets an overall idea how difficult or easy the task was.

I also would not refer to the resonance strategy as null hypothesis, because null hypothesis denotes the claim that the effect being studied does not exist or when there is no relationship between two conditions.

L 116: Which statistical comparison was applied? A linear mixed model with the two oscillations or did you use t-tests for each of the two comparisons? If yes, them report t statistics with degrees of freedom and Bonferroni adjustments. See comment on statistics. Table 1: use the same labels for the pendulums.

Line 131: Which data were entered to calculate the means and standard deviations? All trials and all s subjects? Again, report the statistical results.

Why were two mode metrics calculated, one for position and one for the full state space? The results are very similar and only one of the two is needed. I suggest the more complete metric for the full state space. Figure 2 does not have red traces?

Line 139-146: The statistical evaluations for these important tests are obscure and the conclusion that there is no significant difference, although P90 does differ, is sketchy. There are two significant p-values, which suggest that participants’ dynamics are somehow different from BS1 (in position for P0 and over the full manifold for P90). While participants are closer to BS1 than to the other strategies, they still differ when using this analysis. Suggestion: compare the subjects’ data (include 3 trials as separate estimates) using a linear mixed model with the 3 hypothesized strategies. The mode metric is likely to be lowest for BS1 as hypothesized.

Table 1: Comparing values of BS1 and Exp1, it seems that participants were able to perform the task “better” than BS1 based on the metrics introduced. Could this suggest that even BS1 is not the correct baseline strategy to compare against participant strategy? I think it is important for the authors to touch on this in the discussion.

Figure 2: target locations are ellipses indicating different axes. Better make axes the same +1 to -1m to give a more realistic impression. May be the same for joint space depiction.

The section on applied handle motion is hard to follow as its rationale was not introduced or prepared in any way. See more detailed comments in the following:

Line 155-157: Sentence ‘Therefore …require a control input’ is unclear. I understand that if the handle is displaced, the conservative system would continue with the handle moving accordingly. Would this not imply that the larger handle deflection and sinusoidal motion with the initial amplitude needs to continue? Alternatively, I understand that at resonance the input amplitude can be small. In this case, the friction in the pendulum requires continued input. These three observations do not quite converge.

Line 160: If 1.55 rad is the amplitude, then it would be better to say that the amplitude of P0 was 1.5 rad, not moved more … I do not see the numbers how this amounts to 14% or 17% larger amplitude. Is there an explanation for the offset of the human strategy compared to the simulated strategy? Also, these values are 0.5pi, as stated below. Relate this number to .5pi before when the numbers are presented.

Line 179: past tense is confusing here, the phase will be compared to task success (task success should be mentioned first in the results).

Lines 179-192: correlations were illustrated in Figure 6 with participants ranked by either the two score metrics with a linear trend line, paired with values of the phase lag highlighted by a separate linear trend line. This is a confusing way to report correlation. Illustrate correlation analysis via scatter plots of X vs. Y. Consider also adding confidence bands.

Line 188: both correlation coefficient and p-value are relatively far from significance, rephrase.

Line 192: the highest value was reported to be .74pi?

Line 207: replace ‘stretched’ pendulum with ‘extended’ pendulum here and below. Do the reported hit numbers for the participants include both hits and overswing? Please report task success with all three occurrences.

Lines 217-220: If I understand this correctly, in order to become more accurate, participants shifted the oscillation frequency farther away from BS1? Does this not negate the hypothesis of the authors? If participants want to be more accurate, they should more accurately match their oscillation frequency to the “resonant” frequency of the system.

Also, the tests should again be corrected and include 3 trials.

Figures 5, 6 and 7: Axis labels saying position or velocity are redundant. Could be removed to reduce clutter. Also, for the two left panels, y axis could be θ [rad]. What denotes the green color in Figure 5? Caption word over all correct to ‘overall’. Report hit rate consistently as number, not percent.

Line 245: “..a setup is unlikely, ‘such that’ we used an additional..”. ‘such that’ is not English here

Line 275: “..data supporting the ‘intuitive’ hypothesis..” The hypotheses tested here are not purely intuitive but theoretically and quantitatively motivated.

The results of Experiment 2 also warrant a quantitative comparison with the mode metric. As it stands, the results are only reported visually.

Lines 250-270: the results of this experimental condition should again be compared to the three hypothesized strategies, using the three descriptive metrics: target hits, mode metric, and phase lag.

Discussion

Line 304: the fact that human handle trajectories were close to sinusoidal, similar to the sinusoidal input of BS1 was not really dealt with and tested in the results. If this is a strong observation, this should be tested.

Lines 310-322: In the rest of the manuscript, the “training” period was completely omitted in the analysis. Yet, data from that period are discussed here as though the reader has been previously presented some of the trends and behaviors discussed. For example, on line 318 it is mentioned “the learning curve during the training period appeared very steep”. If this is an important point that needs to be discussed, then perhaps a figure should be added to show the mentioned learning curve. The way the text stands now, it feels very vague and unsubstantiated.

Line 327: The statement that subjects adopted a prospective control approach appears bold as it is only based on the assumption that resonance behavior is more predictable as reported in another study. Such relations need to be reported more carefully as conjectures, rather than facts.

Lines 310-342: this is quite lengthy and could be made more succinct.

Line 348: Could I be that the close location of the handle on the screen biased the subject as otherwise there may be an overlap?

Line 359: rephrase ‘dedicated’

Line 365: this discussion is confusing: it is proposed that the visual display biased subjects and that this then causes deviations from the sinusoidal trajectory shape. This contradicts what was said above.

Line 398-411: these statements about chaotic motion and control are confusing. First, the human data were not analyzed whether they revealed chaotic features. They were only compared to the NNM. The discussion about chaotic control or predicting chaotic evolution appear disconnected from the results.

Line 413-432: I am not convinced by the argument made in this section explaining why participants slightly reduced their movement frequency when the target size was reduced. Fitt’s Law does not seem like a valid argument to be made here. While it is true that humans often obey the speed-accuracy trade-off, this is not always the case as conflicting constraints may suppress it: Kelso, Southard, Goodman, 1979). But if speed-accuracy trade-off is the primary driving factor, then BS2 must be the best strategy as it is the slowest, hence the most accurate strategy.

Line 457: sentence grammar ‘it is not only ..

Line 460: Here resonance sensitivity and prospective control is equated. This is not correct.

Methods

Figure 8a: increase the turning points (hollow dots) that are hardly visible to ease understanding of the figure.

Figure caption: delete ‘a’ in line 2, change second d) to e)

Line 541: exemplary, … not English, change to: As an example,..

Also, it is nearly impossible to see these dotted lines on the figure, if they are important to be pointed out, then perhaps use a different color to plot them.

Figure 9, caption: joint space is depicted on the right, not left

Line 561: “level E =2.5 J that is arbitrarily chosen”

In the previous version of the manuscript, the justification for this choice of energy level was based on “human preference”. Knowing that, to say a value was arbitrarily chosen seems like the authors did not want to spend an extra sentence giving a deeper and more satisfying justification for this choice.

Lines 561-562: “In this investigation, only the first, more stable mode will be considered”

How was ‘more stable’ quantified and identified?

Line 581: “two links with ball geometry were added as targets that had to be reached.” I do not understand whether the ball geometry refers to the links or the targets, rephrase

Line 583: This is not a collision with force, change collision to the link reached or hit the target

Lines 598-600: “In the case of the random control, the amplitude A and frequency ω were pseudo-randomly resampled every 0.1 s) with A ∼ U (1, 10) and ω ∼ U (0, 10) rad s−1.”

For A ∼ U (1, 10), does this mean that the amplitude of oscillation could be 10 radians? Which means multiple complete rotations (2pi) of the link?

Figure 12: While these figures now make understanding the task easier, I wonder what the hand motion was: wrist flexion and extension in the general position shown in Figure 12c? Or was it a forward/backward movement? How did this match with the visual display of the red handle?

The experiment figure would be better combined as the initial figure in the introduction.

Line 675: “this specific parameter” there is no parameter value mentioned yet. I suppose you mean the value mentioned in the next sentence? Also, the sentence does not give a real rationale: if humans add their own stiffness, why do you need a stiffness in the virtual motor link?

Line 704-705: “After one completed experiment, the participants filled out a NASA TLX-inspired questionnaire.” Why include this if the questionnaire is not discussed anywhere else in the manuscript?

Line 706: I would rephrase baseline strategies to hypothesized strategy and alternative strategies, HS and AS1 and AS2. Or give is short names. To label strategies BS is too close to slang for ‘bull shit’ (sorry).

Line 725: a reference for ‘humans seek predictability’ is Nasseroleslami, Hasson, Sternad (2014), which first quantifies predictability as an objective for object interactions.

Line 733: I would refrain from citing papers in un-reviewed publication outlets, like Studies in Perception and Action, which is essentially an abstract book. See also Russell & Sternad (2001) for intermittent control or synchronization in slower oscillations and Wolpert, Miall et al (1992) Evidence for an error deadzone (1992) or Park, Marino, Charles et al (2017) Moving slowly...

Lines 745-758: Description of BS3 Control Strategy. It appears that BS3 is an inspired or “softened” form of a true bang-bang controller, in order to make it more feasible for humans to employ. More explanation of BS3 needs to be included to explain how this version is different from a true bang-bang controller with discrete on-off transitions.

**Have the authors made all data and (if applicable) computational code underlying the findings in their manuscript fully available?**

Reviewer #1: Yes

Reviewer #2: **No: **

PLOS authors have the option to publish the peer review history of their article (what does this mean?). If published, this will include your full peer review and any attached files.

Reviewer #1: No

Reviewer #2: No
---

## [Decision Letter · Decision Letter 2]

21 Jun 2024

Dear Ms. Schmidt,

Thank you very much for submitting your manuscript "Finding the rhythm: Humans exploit nonlinear intrinsic dynamics of compliant systems in periodic interaction tasks" for consideration at PLOS Computational Biology. As with all papers reviewed by the journal, your manuscript was reviewed by members of the editorial board and by several independent reviewers. The reviewers appreciated the attention to an important topic. Based on the reviews, we are likely to accept this manuscript for publication, providing that you modify the manuscript according to the review recommendations.

I am going to ask for another revision, but if the reply to the reviewers is sufficiently detailed, I will not send it out for further review. What I would like to see is (1) all the typos should be fixed. (2) I find the paper readable, so how much you rewrite the text is up to you.

Additionally, I will quote here another editor regarding the statistics:

"There's no reason to do an ANOVA with posthoc Tukey if your hypotheses involve individual t tests, you can go on with these, reduce the fishing expedition vibe, and increase the power. Also, correction for multiple comparison is also necessary for multilevel exploratory ANOVA (https://link.springer.com/article/10.3758/s13423-015-0913-5).

The fact that the authors say: "we don't want to apply Bonferroni since this would even favor our own hypothesis" is not accurate, the correction to the alpha level is for evidence against the data being distributed according to the null hypothesis, regardless whose hypothesis this is. Now, frequencies are not independent, the spectrum is somewhat a continuum, so Bonferroni is probably too strict, False Discovery Rate or Benjamini Hochberg would be more appropriate."

Thus, some minor fixes and the paper will be acceptable.

Sincerely,

Bard Ermentrout

Academic Editor

PLOS Computational Biology

Daniele Marinazzo

Section Editor

PLOS Computational Biology

I am going to ask for another revision, but if the reply to the reviewers is sufficiently detailed, I will not send it out for further review. What I would like to see is (1) all the typos should be fixed. (2) I find the paper readable, so how much you rewrite the text is up to you.

Additionally, I will quote here another editor regarding the statistics:

"There's no reason to do an ANOVA with posthoc Tukey if your hypotheses involve individual t tests, you can go on with these, reduce the fishing expedition vibe, and increase the power. Also, correction for multiple comparison is also necessary for multilevel exploratory ANOVA (https://link.springer.com/article/10.3758/s13423-015-0913-5).

The fact that the authors say: "we don't want to apply Bonferroni since this would even favor our own hypothesis" is meaningless, the correction to the alpha level is for evidence against the data being distributed according to the null hypothesis, regardless whose hypothesis this is. Now, frequencies are not independent, the spectrum is somewhat a continuum, so Bonferroni is probably too strict, False Discovery Rate or Benjamini Hochberg would be more appropriate."

Thus, some minor fixes and the paper will be acceptable.

Reviewer's Responses to Questions

**Comments to the Authors:**

Reviewer #2: I applaud the authors for their efforts to address our admittedly very extensive comments. The manuscript is vastly improved compared to the last 2 versions. And yet, I wanted to provide suggestions on several more points detailed below. Especially the statistical analyses are still in need of improvement. Below are my remaining suggestions.

Major Comments:

I still have an issue with the statistics as there is a multitude of t-tests, with or without corrections. Reporting of a multitude of t-tests is regarded as scientifically clumsy and prone to false conclusions. 1) I understand that Reviewer 1 suggested a distinction between a null hypothesis and alternative hypotheses. However, after much discussion with my group and colleagues, I strongly believe that this distinction is not suitable here. The authors want to test three baseline strategies and the NNM for how they account for the data. The expectation is that BL1 will ‘win’ in that there will be no significant differences between the human data and the BL1. Despite this expectation, the aim of testing a hypothesis is to falsify it. If it cannot be falsified, while others can be falsified, here BL1, BL2 and BL3, then this is regarded support for the NNM hypothesis. I think that each of the three BL1, BL2, and BL3 hypotheses should be put to test equally. 2) Bonferroni corrections, if applied to few targeted pairwise comparisons is acceptable. However, your argument that these corrections should not be applied to all tests is not correct to my knowledge based on my first point. 3) A simple solution to this testing issue is to do the following: for each of the four metrics conduct a one-way ANOVA for the subject values minus the values of BL1, BL2, and BL3. If there is a significant main effect, this is a first result stating that the three strategies are different from each other. Then, conduct posthoc pairwise comparisons (using Tukey corrections applied automatically in the statistics packages) to identify whether all pairs are different from each other. Then, perform a separate t-test to evaluate whether the subject data minus BL1 is different from zero.

But please consider a perennial conundrum: why are NNM and BL1 considered separately, since BL1 is derived from the NNM?

The design of the experiments remains unclear: In line 125, the sentence says that each trial is 20s long, later it says 40s. It also states that all conditions are randomized. This cannot be correct. Please state what was done in Experiment 1, e.g., 4 trials each for P0 and P90, randomly shuffled. Then in Experiment 2: ..,, Experiment 3 … Also, were participants informed of pendulum change between experiments? What were they told?

The correlations shown in Figure 4 are atypical as they include participant as a factor that is not needed for the questions asked: Is there a correlation between phase lag and number of hits, and phase lag and the mode metric? The straightforward way to show the data is to directly plot phase lag against hits and phase lag against the mode metric. In each of the two plots you can include the data for P0 and P90 in different colors.

Experiment 3 receives very little coverage. This is odd as it is a very important experiment that shows that also targets not matching the NNM can be hit. This important demonstration should be signaled better. The plausible conclusion is that wrist stiffness is adjusted to change the system behavior to match the new targets. It is only non-optimal that this human-added stiffness is tweaked into the simulations by changing k1. This does not correspond to the human-added stiffness. It is too much to extend the model, but this ‘quick fix’ should be commented on. In principle, it appears better to me to have used the same pendulum configurations and only changed one thing, the target locations

The labeling is sometimes odd. For example, reference to the upper link and upper spring of the pendulum is misleading as the pendulum is not suspended in gravity and the first link points to the left.

Mention that approximating the human stiffness with a change of k1 is not entirely correct, but a model extension would extend the scope of the paper.

The experimental set-up is clear now. But for future studies, I wanted to make one comment: in the virtual implementation, the handle overlies the double pendulum, which in this constellation is physically impossible to manipulate. The hand and the upper first link of the pendulum coincide. I would regard much more intuitive of the handle was non-overlapping and pointing to the right of the system.

The descriptions are still very wordy. In addition, there is a lot of duplication between the results and the methods. In this order of results and methods (which I think is not helpful), the result section needs some wording to make the results understandable, including the motivation of experimental and analysis choices. The methods should simply describe the methods without any further embedding into the scientific rationale. In the current version, the methods contain all the motivation and reasoning, which makes them overly long and repetitive. For example, the BL3 strategy should simply be described: a bang-bang controller was used of the following form .. eqs 8 and 9.

Abstract:

Last sentence not needed. This manuscript is not introducing NNMs. Therefore, this sentence is a misrepresentation of the current contributions and must be removed.

Instead, I would mention that you conducted 3 experiment and Experiment 3 demonstrated that subjects could also hit target off the NNM, probably by stiffening their wrist.

Introduction

“To verify if the excited motions match the actual intrinsic dynamics of the objects, nonlinear dynamics need to be computable for comparison.” The English is not correct here.

Line 12: “..bouncing a ball by supporting intrinsic system dynamics” Supporting is the wrong word here.

Line 42: “This uniquely enables us to..” The word uniquely sounds odd.

Lines 189-191: Is the conversion in degrees needed?

Line 204: “..which is compared to the baseline strategies again closest to the values obtained for BL1.” The word again here is wrong. The whole sentence structure is wrong but I am not sure exactly what the authors want to convey.

The values for the mode metric has no defined scale, hence the absolute values are not meaningful. It may therefore be good to compare the values against each other with statistical comparisons.

Line 440: The reference to the information bandwidth as in Fitt’s Law is unclear and not needed.

Line 589. Indicate the recording frequency

Line 601. A parenthesis is lacking ‘(‘, and there is no unit of measure after the range of A.

Line 622: what is a high-score game?

Line 652: group, r remained … delete ‘the’

Line 684: “following’ .. delete

Line 694-701: very confusing description. It remains unclear whether each trial was from a different condition, or blocked. Exp 3 has the different targets? Why coupled with Exp 1? There must have been considerable breaks to change the parameters for Exp2?

Line 723: tested the investigated systems .. delete ‘on’

Line 728-743: lengthy complex justification for BL2 which is a simple controller at half of the eigenfrequency: reliant on feedback and smooth, low effort, I find the three considerations confusing and conflicting, e.g. slower frequency is not necessarily low effort, and relying on feedback is not necessarily smooth.

Line 739. “The amplitude was once more tuned empirically to hit the targets.” The authors should indicate the amplitude values.

Line 776-780: this allocation of points including subtraction of points contradicts what was written higher up in the results.

Line 787: delete ‘attempts’ in error attempts

Line 855: this is problematic: a statistical test aims to falsify, to not show a difference can be due to noise in the data. The argument for omitting Bonferroni corrections is therefore sketchy. See above for thoughts on a simple analysis strategy.

**Have the authors made all data and (if applicable) computational code underlying the findings in their manuscript fully available?**

Reviewer #2: Yes

PLOS authors have the option to publish the peer review history of their article (what does this mean?). If published, this will include your full peer review and any attached files.

Reviewer #2: No

Figure Files:

Data Requirements:

Reproducibility:

References:

---

## [Editor Report · Decision Letter 3]

8 Aug 2024

Dear Ms. Schmidt,

We are pleased to inform you that your manuscript 'Finding the rhythm: Humans exploit nonlinear intrinsic dynamics of compliant systems in periodic interaction tasks' has been provisionally accepted for publication in PLOS Computational Biology.

Best regards,

Bard Ermentrout

Academic Editor

PLOS Computational Biology

Daniele Marinazzo

Section Editor

PLOS Computational Biology

---

## [Editor Report · Acceptance letter]

26 Aug 2024

PCOMPBIOL-D-23-01396R3 

Finding the rhythm: Humans exploit nonlinear intrinsic dynamics of compliant systems in periodic interaction tasks

Dear Dr Schmidt,

I am pleased to inform you that your manuscript has been formally accepted for publication in PLOS Computational Biology. Your manuscript is now with our production department and you will be notified of the publication date in due course.

With kind regards,

Anita Estes
